# Understanding the genetic complexity of puberty timing across the allele frequency spectrum

Pubertal timing varies considerably and is associated with later health outcomes. We performed multi-ancestry genetic analyses on ~800,000 women, identifying 1,080 signals for age at menarche. Collectively, these explained 11% of trait variance in an independent sample. Women at the top and bottom 1% of polygenic risk exhibited ~11 and ~14-fold higher risks of delayed and precocious puberty, respectively. We identified several genes harboring rare loss-of-function variants in ~200,000 women, including variants in *ZNF483*, which abolished the impact of polygenic risk. Variant-to-gene mapping approaches and mouse gonadotropin-releasing hormone neuron RNA sequencing implicated 665 genes, including an uncharacterized G-protein-coupled receptor, *GPR83*, which amplified the signaling of *MC3R*, a key nutritional sensor. Shared signals with menopause timing at genes involved in DNA damage response suggest that the ovarian reserve might signal centrally to trigger puberty. We also highlight body size-dependent and independent mechanisms that potentially link reproductive timing to later life disease.

Age at menarche (AAM), the onset of menses in females, represents the start of reproductive maturity and is a widely reported marker of pubertal timing. Menarche normally occurs between ages 10 and 15 years[1], and its variation is associated with risks of several health outcomes, including obesity, type 2 diabetes, cardiovascular disease and hormone-sensitive cancers[2–6]. Thus, widespread secular trends toward earlier puberty timing may have an important impact on public health[7]. AAM is a highly polygenic trait[8], and previous genome-wide association studies (GWAS) have identified ~400 common genetic loci[9–13], the vast majority of which were discovered in samples of European ancestry. AAM has a strong genetic correlation with male puberty timing ($r_g = 0.68$)[14], as well as with adiposity (body mass index (BMI), $r_g = -0.35$)[9], and specific pathways have been identified that link nutrient sensing to reproductive hormone axis activation. For example, we recently reported that *MC3R* is the key hypothalamic sensor linking nutritional status to puberty timing[15].

Previously reported GWAS signals in ~370,000 women of European ancestry explained ~7.4% of the population variance in AAM,

corresponding to ~25% of the estimated heritability[9]. Here through an expanded GWAS in 799,845 women, including 166,890 of East Asian ancestry, we identify 1,080 independent signals for AAM. Female participants carrying an excess of these alleles have the equivalent risk of precocious or delayed puberty compared to those carrying clinically relevant monogenic alleles. We complement these common variant analyses by undertaking the first large-scale assessment of rare variation in puberty timing in 222,283 women with exome sequence data. Through subsequent variant-to-gene mapping approaches, we implicate 665 genes, which collectively shed further light on the biological determinants of puberty timing and the mechanisms linking it to disease risks.

## Results

We performed a GWAS meta-analysis for AAM in up to 799,845 women by combining data from the following five strata: (1) 38 ReproGen consortium cohorts ($n = 180,269$), (2) UK Biobank ($n = 238,040$), (3) the Breast Cancer Association Consortium and the Ovarian Cancer

✉ e-mail: john.perry@mrc-epid.cam.ac.uk

Association Consortium ($n$ = 137,815), (4) 23andMe ($n$ = 76,831) and (5) three East Asian biobanks—the China Kadoorie Biobank, the Biobank Japan and the Korean Genome and Epidemiology Study ($n$ = 166,890). Studies in strata one to four comprised European-ancestry individuals. All studies provided GWAS data imputed to at least 1000 Genomes reference panel density (Supplementary Table 1), yielding a total of ~12.7 million genetic variants in the final meta-analysis. We did not find evidence of test statistic inflation due to population structure (linkage disequilibrium score regression (LDSC) intercept = 1.07, s.e. = 0.03).

To maximize the discovery of AAM genomic signals, we used a combination of distance-based clumping and approximate conditional analysis (Methods) in the European-strata and ancestry-combined meta-analyses, to identify signals that are homogenous across the two ancestry groups. European-strata identified signals ($n$ = 935) were supplemented with additional signals from the ancestry-combined analysis ($n$ = 145), resulting in a total of 1,080 statistically independent signals for AAM at genome-wide significance ($P < 5 \times 10^{-8}$; Fig. 1 and Supplementary Table 2). Effect sizes ranged from 3.5 months per allele for rarer alleles (minor allele frequency (MAF) = 0.9%) to ~5 days per allele for more common variants (Supplementary Fig. 1). Across the 145 additional signals, we observed a median 1.16-fold increase in $\chi^2$ for their association with AAM in the ancestry-combined analysis compared to European-only, which is proportionate to the added number of East Asian samples (~21% of the total).

Independent replication data from the Danish Blood Donor Study (DBDS; $n$ = 35,467; Supplementary Table 3) was available for 969/1,080 signals[16]. Of these, 862 showed directionally concordant associations (89%, $P_{binomial}$ = 2.9 × 10⁻¹⁴⁷). In this independent sample, the variance explained in AAM doubled from 5.6% for 355 available previously reported signals[9] to 11% for the 969 available signals. We also sought indirect confirmation of AAM signals by association with age at voice breaking (AVB) in men from the UK Biobank study ($n$ = 191,235) and 23andMe ($n$ = 55,871; Supplementary Table 4)[14,17,18]. Of the 1,080 AAM signals, 909 (84%, $P_{binomial}$ = 2.6 × 10⁻¹²²) showed directionally concordant associations with AVB in UK Biobank (including 354 at $P < 0.05$). Similarly, 852/1,067 (79%, $P_{binomial}$ = 1.8 × 10⁻⁹⁰) AAM signals available in 23andMe showed directionally concordant associations with AVB (217 at $P < 0.05$). In the combined dataset (effective $n$ = 205,354), 893/1,020 (83%, $P_{binomial}$ = 1.18 × 10⁻¹⁴²) signals showed directionally concordant effects (397 signals at $P < 0.05$).

**Exome analyses identify new rare variants of large effect**

Previous genetic studies for AAM have largely been restricted to assessing common, largely noncoding, genetic variation. We sought to address this by performing an exome-wide association study (ExWAS) on 222,283 European-ancestry women in the UK Biobank. Gene burden tests were performed by collapsing rare variants (MAF < 0.1%) in each gene according to the following two overlapping predicted functional categories: (1) high-confidence protein truncating variants (HC PTVs) and (2) HC PTVs plus missense variants with CADD score ≥25 (ref. 19; termed 'damaging variants' (DMG)). Six genes were associated with AAM at exome-wide significance ($P < 1.54 \times 10^{-6}$, 0.05/32,434 tests; Fig. 2, Supplementary Figs. 2–4 and Supplementary Table 5). This included the following two genes previously reported in rare monogenic disorders of puberty: *TACR3* ($\beta$ = 0.62 years, $P$ = 3.2 × 10⁻¹⁹, $n$ = 489 DMG carriers) previously implicated in normosmic idiopathic hypogonadotropic hypogonadism (IHH)[20] and *MKRN3* ($\beta$ = −0.59 years, $P$ = 1.4 × 10⁻⁷, $n$ = 187 DMG carriers) previously implicated in familial central precocious puberty[21]. Furthermore, *MC3R* ($\beta$ = 0.33 years, $P$ = 1.6 × 10⁻⁹, $n$ = 796 DMG carriers) was recently reported to link nutrient sensing to key hypothalamic neurons[15].

Of the three new genes, *KDM4C* ($\beta$ = −0.33 years, $P$ = 2.5 × 10⁻⁷, $n$ = 582 DMG carriers) encodes a lysine-specific histone demethylase likely involved in epigenetically regulating hypothalamic–pituitary–gonadal (HPG) axis genes[22]. In addition, *PDE10A* ($\beta$ = 0.58 years,

$P$ = 1.2 × 10⁻⁷, $n$ = 196 DMG carriers) encodes phosphodiesterase 10A, which regulates the intracellular concentration of cyclic nucleotides and hence signal transduction[23]. Finally, *ZNF483* ($\beta$ = 1.31 years, $P$ = 4.9 × 10⁻¹¹, $n$ = 59 DMG carriers) encodes a zinc finger protein transcription factor involved in neuronal differentiation[24] and self-renewal of pluripotent stem cells[25].

We were able to confirm four of these six genes (*KDM4C*, *MC3R*, *TACR3* and *ZNF483*) using voice-breaking data in 178,625 men with exome sequence data in the UK Biobank ($P < 0.05$; Fig. 2 and Supplementary Table 5). Lack of association with AVB at *MKRN3* is consistent with previous reports that rare *MKRN3* mutations have a greater clinical impact in girls than boys[21,26]. None of the genes showed an association with childhood or adult adiposity (Supplementary Table 6).

In addition, we specifically examined rare variant associations with AAM or VB for *ANOS1*, *CHD7*, *FGF8* and *WDR11*, which are clinically tested in hypogonadotropic hypogonadism ('high-evidence genes' on the Genomics England IHH panel[27]) and show a dominant or X-linked mode of inheritance (Supplementary Table 7). Normal puberty timing (AAM: 10–15 years[1] or VB: 'about average') was reported by all carriers of PTVs in *ANOS1* ($n$ = 5 male) and *CHD7* ($n$ = 5 female and $n$ = 1 male). PTVs in *WDR11* showed no association with delayed puberty, with only 7/81 female and 5/68 male carriers reporting delayed puberty. Female carriers of PTVs in *FGF8* showed some evidence of later puberty ($\beta$ = 1.4 years, $P$ = 3.6 × 10⁻³, $n$ = 5/10 reported delayed puberty) but with no effect in males ($n$ = 1/8 reported delayed puberty; Supplementary Fig. 5). These observations highlight the lower penetrance of rare deleterious variants in large population-based studies compared to patient cohorts[28–30].

**Common variation influences the risk of phenotypic extremes**

Rare pathogenic variants, such as the ones mentioned above, are described to cause disorders of puberty. However, it remains unclear whether common genetic variants also contribute to abnormal puberty timing. To assess this, we generated a polygenic score (PGS) of AAM in a penalized regression framework using lassosum[31] and data from our meta-analysis of European-ancestry cohorts but excluding the UK Biobank. This PGS explained ~12% of the phenotypic variance in UK Biobank. The PGS was informative in individuals experiencing menarche as early as 8 years old and later than 20, well beyond the normal AAM range (Supplementary Fig. 6 and Supplementary Tables 8 and 9).

We next sought to understand how the risks of early (<10 years) and delayed (>15 years) AAM were influenced by the PGS. Women in the lowest PGS centile reported AAM at 11.49 (mean, s.e. = 0.03) years, compared to 14.46 (0.04) years in the top 1% (Supplementary Fig. 6 and Supplementary Table 10). Compared to women in the 50th PGS centile, those in the top 1% PGS were 10.7 times more likely to report late AAM (odds ratio (OR) = 8.20–13.96, $P$ = 2.6 × 10⁻⁶⁸), while women in the lowest 1% were 14.2 times more likely to report early AAM (OR = 7.13–28.39, $P$ = 5.1 × 10⁻¹⁴). Collectively, these findings suggest that common genetic variants contribute to the risk of rare clinical disorders of extremely early (precocious) and delayed puberty.

To evaluate the predictive performance of our AAM signals, we compared them to phenotypic predictors in 3,140 female children from the Avon Longitudinal Study of Parents and Children (ALSPAC) study. The AAM signals in combination explained more variance in AAM than childhood BMI, parental BMI or mother's AAM (Supplementary Table 11). Furthermore, they had a similar ability to predict extremes of AAM (beyond 2 s.d.) than a multiphenotype predictor (Supplementary Fig. 7), and a combined genotype and phenotype model showed high predictive ability for early (area under the receiver operating characteristic curve (AUROC) = 0.75 (95% confidence interval (CI), 0.68–0.82)) and late AAM (AUROC = 0.85 (95% CI, 0.81–0.92)).

We next tested whether carrying rare variants in the AAM ExWAS genes modifies the common polygenic influence on AAM. We saw that the effect of the common variant PGS on AAM was attenuated in the

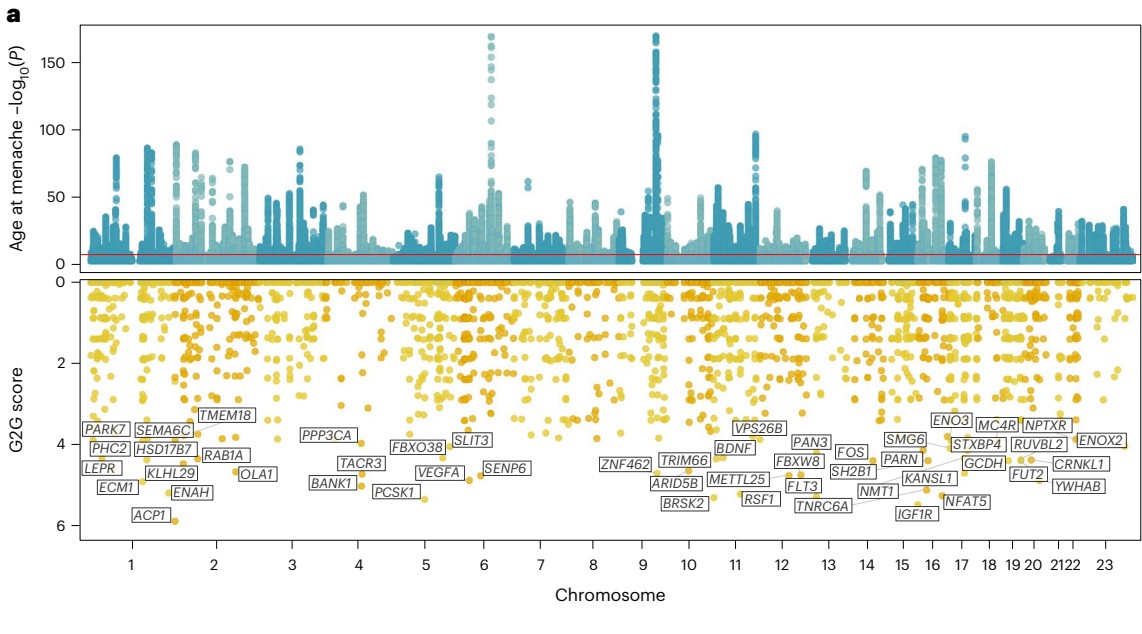

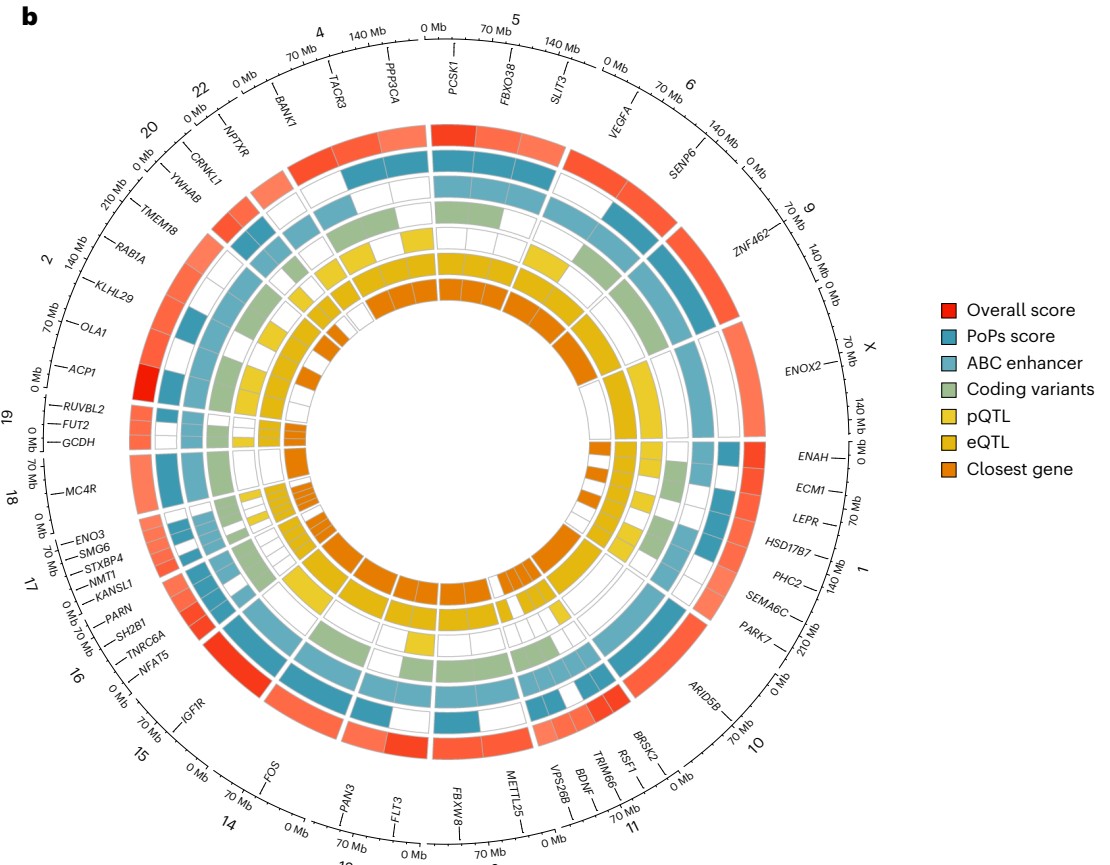

**Fig. 1 | AAM GWAS and gene prioritization. a**, Miami plot showing signals from the European meta-analysis for AAM (top), and G2G scores with names of the top 50 genes annotated (bottom). Top, *y* axis is capped at $-\log_{10}(1 \times 10^{-150})$ for visibility. **b**, The 50 top-scoring genes implicated by G2G, annotated by their sources of evidence. Relevant data are included in Supplementary Tables 2 and 13–16.

49 unrelated carriers of DMG variants in *ZNF483* ($\beta_{noncarriers}$ = 0.564 years per s.d., s.e. = 0.003; $\beta_{carriers}$ = 0.084, s.e. = 0.214; $P_{interaction}$ = 0.025; Fig. 3 and Supplementary Table 12). To confirm that this was not a reflection of reduced power due to the low number of carriers, we estimated the expected relationship for noncarriers in 10,000 random subsamples of 49 participants and found that the observed carrier effect was unlikely by chance (*P* = 0.015; Supplementary Fig. 8).

Using ENCODE chromatin immunoprecipitation followed by sequencing (ChIP–seq) data[32], we found that the transcriptional targets of *ZNF483* are enriched for in the AAM GWAS (functional GWAS (fGWAS)[32]; *P* = 2.6 × 10⁻⁷) and that greater *ZNF483* binding confers earlier AAM (signed linkage disequilibrium (LD) profile regression (SLDP)[33]; *Z* = −4.9, *P* = 4.8 × 10⁻⁷), which is directionally concordant with the observed effect of rare DMG variants. This was further corroborated

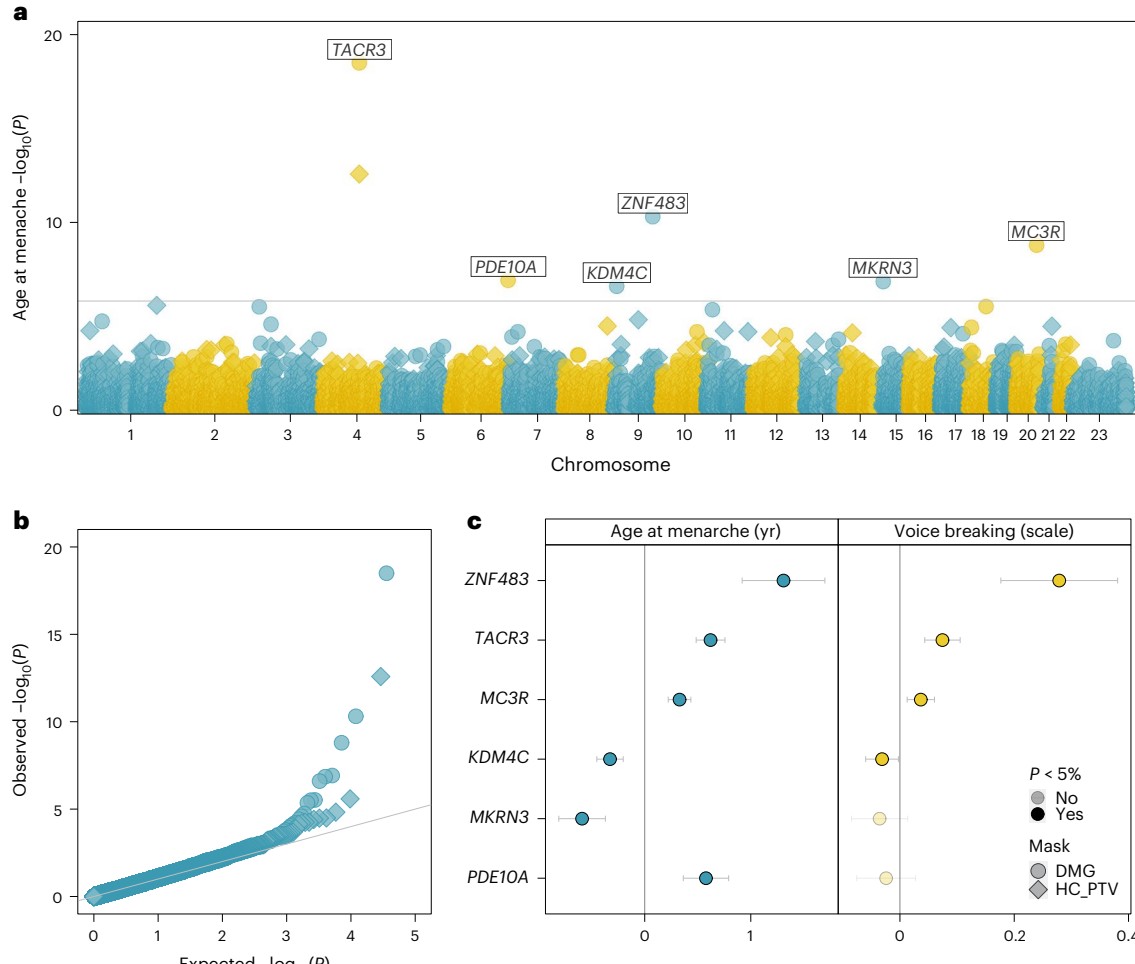

**Fig. 2 | Exome-wide rare variant burden associations with AAM. a**, Manhattan plot showing gene burden test results for AAM. Six genes passing exome-wide significance ($P < 1.54 \times 10^{-6}$) are highlighted. Point shapes indicate variant-predicted functional class (DMG; damaging). **b**, Quantile–quantile (QQ) plot for gene burden tests. **c**, Comparison of gene burden associations (effect sizes with 95% CIs) for AAM (female participants, in years, $n = 222,283$) and AVB (male participants, in three categories, $n = 178,625$). Relevant data are included in Supplementary Table 5.

by functional-domain-specific gene burden analyses, which showed a larger effect on AAM of *ZNF483* DMG variants located within zinc finger domains ($\beta = 1.615$ years, s.e. $= 0.293$, $P = 3.59 \times 10^{-8}$) than DMG variants outwith these domains ($\beta = 0.816$ years, s.e. $= 0.298$, $P = 6.2 \times 10^{-3}$; Fig. 3). This data suggest that rare DMG variants in *ZNF483* confer later AAM by disrupting the protein's ability to bind to its downstream targets.

**Implicating AAM genes through variant-to-gene mapping**

To implicate putatively causal genes that underlie our 1,080 common variant signals for AAM, we developed the framework 'GWAS to genes' (G2G) that integrates genomic and functional data across six sources (Methods; Fig. 1 and Supplementary Tables 13–16). We identified proximal genes (within 500 kb upstream or downstream) for the 1,080 AAM signals and scored them based on the degree of evidence linking our associated variants to the function of these genes. To achieve this, we implicated genes by identifying signals that colocalized with (1) known enhancers and regulatory elements[34], (2) nonsynonymous variants, (3) expression quantitative trait loci (eQTL) specifically in tissues enriched for AAM associations (Supplementary Fig. 9 and Supplementary Table 17) and (4) circulating protein QTL (pQTL) from whole blood (Methods). In addition, we integrated gene-level associations for aggregated nonsynonymous common variants using MAGMA[35] and gene scores from polygenic priority score (PoPs)[36], which uses bulk human and mouse data with information on scRNA, gene pathways and

protein interactions to link genes to GWAS signals. Individual genes were further upweighted if they were the nearest gene to a signal[37,38].

Using this approach, our 1,080 signals were found to be proximal to 10,323 genes, of which 665 'high-confidence AAM genes' were identified as the highest-scoring gene at a locus and with at least two lines of evidence (Supplementary Fig. 10 and Supplementary Tables 18 and 19). High-confidence AAM genes include established components of the HPG axis that are disrupted in rare monogenic disorders of puberty (*CADM1*, *CHD4*, *CHD7*, *FEZF1*, *GNRH1*, *KISS1*, *SPRY4*, *TAC3*, *TACR3* and *TYRO3*)[39] and other recently reported candidate genes (*PLEKHA5*, *TBX3* and *ZNF462*)[40,41]. Other AAM genes have recognized roles in sex hormone secretion and gametogenesis (*ACVR2A*, *CYP19A1*, *HSD17B7*, *INHBA*, *INHBB*, *MC3R* and *PCSK2*)[42], are disrupted in rare monogenic disorders of multiple pituitary hormone deficiency (*OTX2*, *SOX2*, *SOX3* and *SST*)[43], monogenic obesity (*BDNF*, *LEPR*, *MC4R*, *NTRK2*, *PCSK1* and *SH2B1*)[44] or syndromes characterized by hypogonadism (Noonan syndrome: *BRAF* and *SOS1*; Bardet–Biedl syndrome: *BBS4*; Prader–Willi/Angelman syndrome: *NDN*, *SNRPN* and *UBE3A*)[45-48]. Other mechanisms implicated by high-confidence AAM genes include insulin and insulin-like growth factor (IGF) signaling (*CALCR*, *GHR*, *IGF1R*, *INSR*, *NEUROD1*, *NSMCE2*, *PAPPA2* and *SOCS2*)[49]; thyroid hormone signaling (*THRB*)[50] and the polycomb silencing complex (*CBX4*, *CTBP2*, *FBRSL1*, *JARID2*, *PHC2*, *SCMH1* and *TNRC6A*)[51]. We also found strong supportive evidence for all genes identified by our ExWAS (Supplementary Fig. 11).

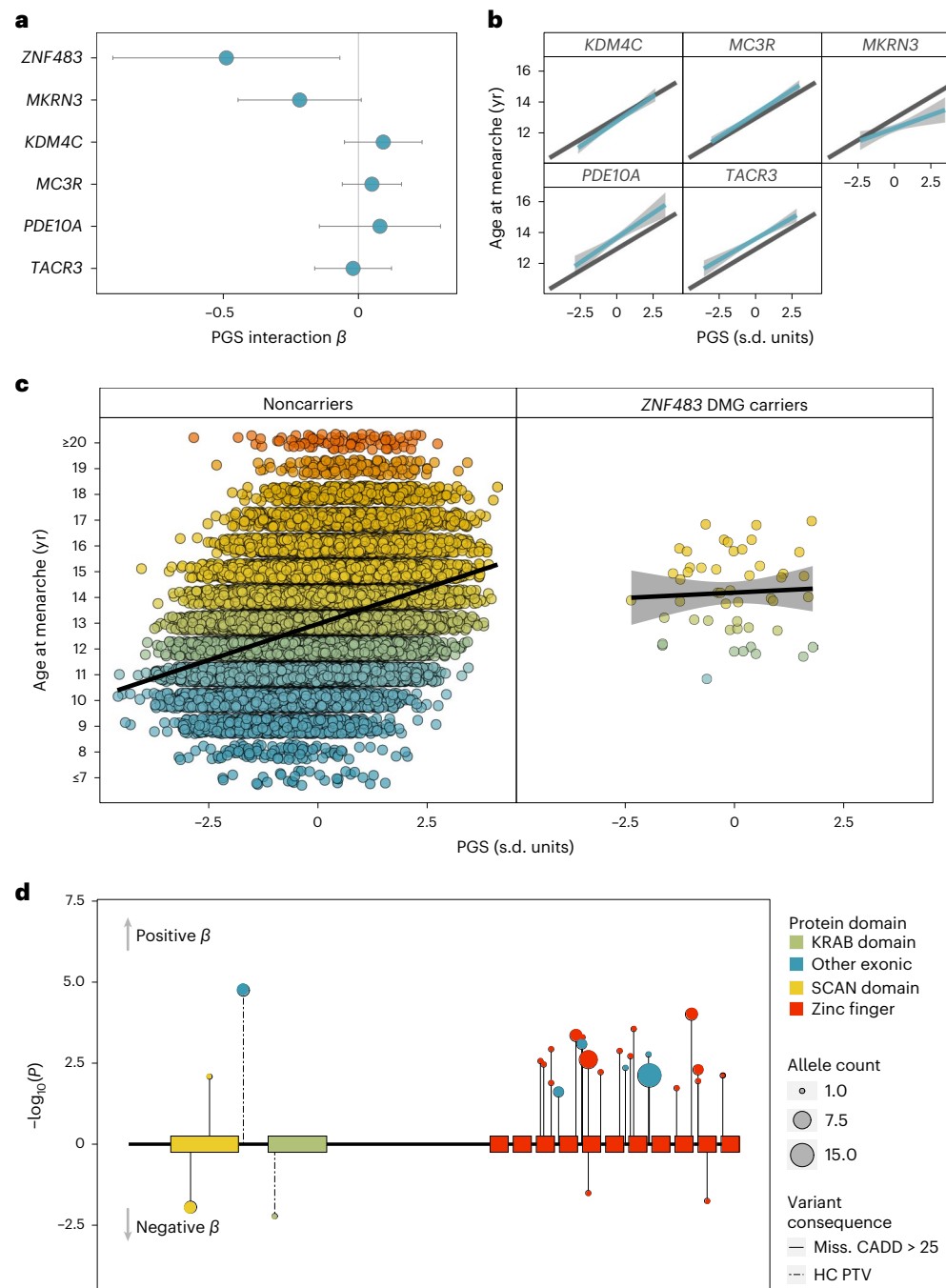

**Fig. 3 | Epistatic interactions between rare coding variants and common genetic susceptibility on AAM in the UK Biobank. a**, Interaction effects (mean and 95% CIs) on AAM between a GWAS PGS and carriage of qualifying rare variants in the six exome-highlighted genes ($n = 222{,}283$). **b,c**, Predicted mean (with 95% CI) AAM in (**b**) noncarriers (black) and carriers (light blue) of rare variants in six genes without significant ($P > 0.05$) interaction effects and (in **c**) in noncarriers (left) and carriers (right) of rare variants in *ZNF483*, which shows significant interaction ($P = 0.025$). In **c**, points show individual AAM values, with the color gradient indicating increasing age. **d**, Plot of individual rare damaging (DMG) variant associations with AAM by *ZNF483* functional domains. The coding part of *ZNF483* is depicted by the horizontal black line. Included DMG variants had an MAF < 0.1% and were annotated to be either HC PTVs or missense variants with CADD score ≥25. Each variant association is represented by a circle and vertical line–the line length indicates the $-\log_{10}(P)$, in the direction of its effect on menarche in carriers of the rare allele, and the circle size indicates the number of carriers of each variant (that is, allele count). Relevant data are included in Supplementary Table 12.

## Weight gain-related and unrelated puberty signals

Phenotypic, genetic and mechanistic links between higher BMI and earlier AAM are well described[15], but it is challenging to distinguish whether individual AAM signals have a primary effect on puberty or weight[9]. Here 83 of the 1,080 AAM signals colocalized (at posterior probability (PP) ≥0.5) and also showed genome-wide significant association with adult BMI (Supplementary Table 20), and 53 further AAM

signals colocalized with adult BMI and showed association with BMI at $P < 4.6 \times 10^{-5}$ (based on 1,080 tests). Of these 136 colocalizing signals, 126 showed an association between the AAM-reducing allele and higher adult BMI (Supplementary Table 20).

To identify AAM signals with or without a primary effect on early life weight gain, we clustered the 1,080 AAM signals by their associations with body weight from birth to age 8 years (before the onset of

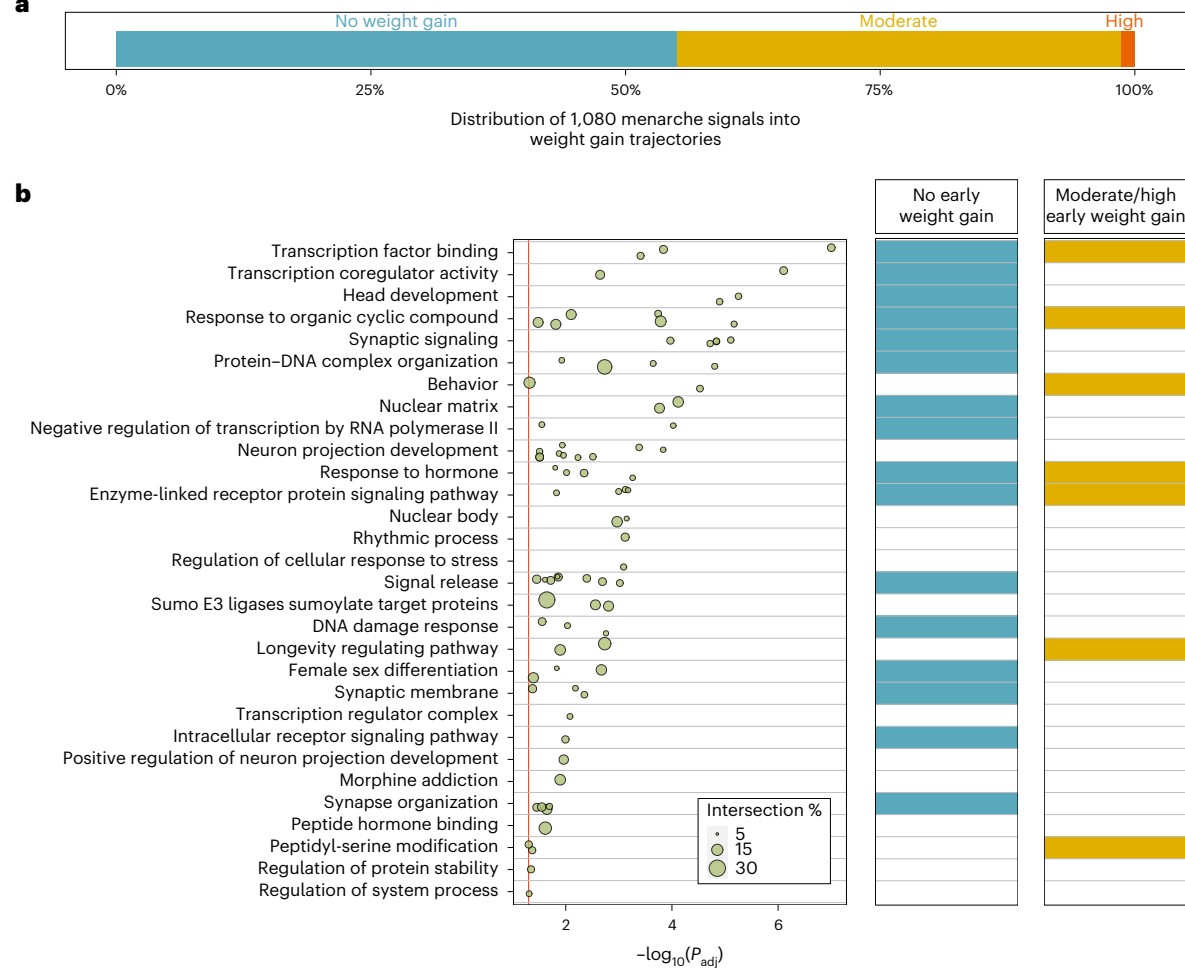

**Fig. 4 | Stratification of AAM signals and biological pathway enrichment by their influence on early childhood weight. a**, Proportion of GWAS signals for AAM by early childhood weight trajectory. **b**, Biological pathways enriched for high-confidence AAM genes (left), plus enrichment within early childhood weight trajectories (right). Pathway enrichment was calculated using g:Profiler, and the displayed *P* values are Bonferroni-corrected. Row names describe pathway clusters. Strength of associations with individual pathways is indicated by circles. Circle size reflects the proportion of pathway genes that are high-confidence AAM genes (in percentage). Right, determining whether each pathway cluster remains enriched for AAM genes when stratified by early childhood weight trajectory. Supporting data are included in Supplementary Tables 21 and 23–26.

puberty) in the Norwegian Mother, Father and Child Cohort Study (MoBa; *n* = 26,681 children)[52]. We identified three trajectories—464 AAM signals (44%) formed a 'moderate early weight gain' trajectory and 15 (1%) formed a 'high early weight gain' trajectory; both trajectories were characterized by effects of AAM-reducing alleles on higher weight gain across early childhood. The remaining 586 (55%) AAM signals formed a 'no early weight gain' trajectory; yet, in combination, AAM-reducing alleles in this trajectory increased adult BMI ($\beta$ = 0.487 kg m$^{-2}$ yr$^{-1}$; $P$ = 1.6 × 10$^{-20}$; Supplementary Tables 21 and 22, Fig. 4 and Supplementary Fig. 12). This data indicate a bidirectional causal relationship between AAM and body size, with greater early weight gain leading to earlier AAM and also earlier AAM leading to higher adult BMI. This approach provides a clear distinction between AAM signals that have primary effects on puberty timing or early weight gain.

**Pathway, tissue and cell-type enrichment of AAM genes**

Genome-wide common variant AAM associations were enriched for genes expressed in several brain regions, and enrichment was highest in the hypothalamus. Outside the brain, we also observed enrichment for genes expressed in the adrenal gland (Supplementary Fig. 9 and Supplementary Table 17).

We next performed gene-based pathway analyses on the 665 high-confidence AAM genes using g:Profiler[53] and identified

85 enriched pathways (Supplementary Table 23), which were grouped into 30 clusters (Fig. 4, Supplementary Fig. 13 and Supplementary Table 24). These included a number of neuroendocrine, sexual development, protein and transcription regulation pathways. To explore distinct biological pathways by early weight trajectories, we stratified the 665 high-confidence AAM genes into 'early weight gain' (*n* = 404) or 'no early weight gain' AAM genes (*n* = 344; Supplementary Table 25). Early weight gain AAM genes specifically highlighted hormone regulation, feeding behavior, AKT phosphorylation targets and peptidyl-serine modification (Supplementary Fig. 14 and Supplementary Table 26). Conversely, the no early weight gain AAM genes highlighted female sex differentiation, negative regulation of transcription by RNA polymerase II, synapse organization and DNA damage response (Supplementary Fig. 15 and Supplementary Table 26). Pathways involved in response to organic compounds, proteins and hormones were enriched across both AAM gene groups (Fig. 4, Supplementary Figs. 13–15 and Supplementary Table 26).

To understand how AAM-associated genes may exert effects on the HPG axis, we explored their expressional dynamics in mouse embryonic gonadotropin-releasing hormone (GnRH) neurons. GnRH neurons are central to reproductive processes and regulate the HPG axis by stimulating the pituitary secretion of follicle-stimulating hormone and luteinizing hormone[54]. During development, they migrate

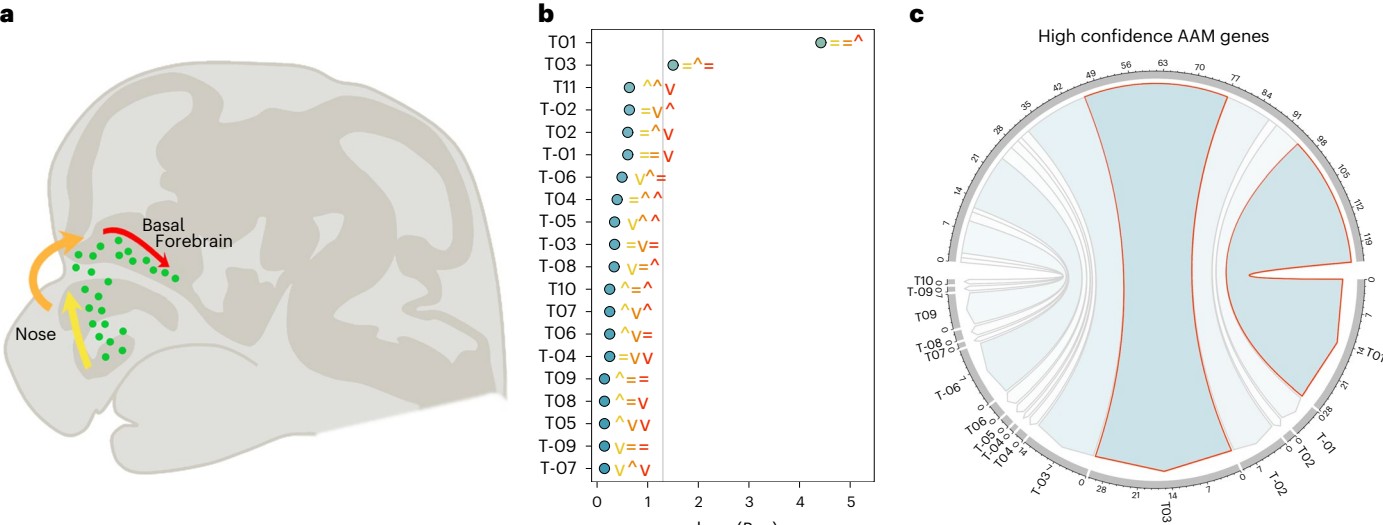

**Fig. 5 | Enrichment of gene drivers of GnRH migration and maturation in the AAM GWAS. a**, Schematic representation of the stages of GnRH neuron migration during embryonic development. Using RNA-seq data, Zouaghi et al.[57] grouped differentially expressed genes into 23 expressional trajectories based on their comparative level of expression during the early (yellow), intermediate (amber) and late (red) stages of GnRH migration. **b**, Genome-wide MAGMA enrichment for AAM gene associations within each expression trajectory. Colored symbols accompanying the points indicate the expression level of genes within each trajectory at the three aforementioned stages of migration; that is, the first yellow symbols indicate whether gene expression remains the same (=), is upregulated (^) or downregulated (v) in the early stage of migration. Orange and red symbols denote the same for the intermediate and late stages of migration, accordingly. **c**, Trajectories significantly ($P_{adj} < 0.05$) enriched at the genome-wide level in **b** are highlighted in red and also show significant ($P < 0.05$) overlap with the 665 high-confidence AAM genes. Supporting data are included in Supplementary Tables 18 and 27.

through the nasal placode and into the hypothalamus, where they establish a neurosecretory network that activates pubertal onset[55,56]. RNA sequencing (RNA-seq) in GnRH neurons previously identified 2,182 genes that showed differential expression between embryonic migration stages (early, intermediate or late; Fig. 5) and were categorized into 23 spatiotemporal expression trajectories[57]. At the genome-wide level, we observed enrichment for AAM associations among genes that become upregulated in the late (Trajectory01, $P_{adj} = 3.8 \times 10^{-5}$) and mid to late stages of GnRH neuron development (Trajectory03, $P_{adj} = 0.032$; Supplementary Table 27), that is, when GnRH neurons have completed their migration process and start to make synaptic connections. Of the 665 high-confidence AAM genes, 28 assign to Trajectory01 ($P_{exact} = 3.2 \times 10^{-6}$), including *NEGR1* and *TNRC6A* and 31 assign to Trajectory03 ($P_{exact} = 7.8 \times 10^{-3}$), including *KDM4C*, *PDE10A* and *TP53BP1*. Both of these GnRH expressional trajectories remained enriched when considering only the subset of nonearly weight affecting genes (Trajectory01, $P_{exact} = 1.8 \times 10^{-5}$; Trajectory03, $P_{exact} = 0.01$), while Trajectory01 was also enriched when considering only AAM genes that influence early weight gain (Trajectory01, $P_{exact} = 1.9 \times 10^{-3}$; Trajectory03, $P_{exact} = 0.09$).

### GPCRs and puberty timing

G-protein-coupled receptors (GPCRs) regulate several endocrine processes and diseases, including puberty timing[58], and are therapeutic targets. Here 24 of 161 brain-expressed GPCRs[59] were implicated in AAM by at least one G2G predictor (Fig. 6 and Supplementary Table 28). These include *MC3R*, where we recently reported that rare loss of function (LOF) variants, which impair signaling, were associated with delayed puberty[15], and *GPR83*, which encodes a Gα_{q11}- and Gα_i-coupled GPCR widely expressed in several brain regions[60,61] and is implicated in energy metabolism[62]. In mice, *Gpr83* and *Mc3r* are reportedly co-expressed in key hypothalamic neurons that control reproduction (kisspeptin, neurokinin B and dynorphin, KNDy neurons) and growth (growth hormone-releasing hormone, GHRH neurons)[15].

Because dimerization between GPCRs may affect their signaling[63], we tested for physical and functional interactions between MC3R and GPR83 in vitro. Using a bioluminescence resonance energy transfer (BRET)-based assay in HEK293 cells, we observed a physical and specific interaction between GPR83 and MC3R (Supplementary Fig. 16 and Supplementary Table 29). We then tested whether GPR83 modifies canonical MC3R signaling by measuring NDP-α-melanocyte-stimulating hormone (NDP-αMSH)-stimulated cyclic AMP production in HEK293 cells following transfection with plasmids encoding wild-type *GPR83* and *MC3R* separately or together (1:1 ratio). *GPR83* and *MC3R* cotransfection increased cAMP production by 43% compared to *MC3R*-alone ($P = 0.03$; Fig. 6, Supplementary Fig. 16 and Supplementary Tables 30 and 31).

Consistent with this in vitro interaction, we observed statistical genetic epistasis between the common AAM signals at *MC3R* rs3746619, a 5′-UTR SNP highly correlated with predicted deleterious coding variants, and *GPR83* rs592068, which colocalizes with eQTLs for *GPR83* in brain[64] and across tissues[65]. Among white European unrelated UK Biobank female participants, *MC3R* function-increasing alleles conferred increasingly earlier AAM in the presence of *GPR83* expression-increasing alleles ($\beta_{interaction} = -0.034 \pm 0.015$ years, $P_{interaction} = 0.02$; Fig. 6). These findings extend our previous observation that *MC3R* loss of function causes delayed puberty[15] and indicate that increased MC3R function through enhanced *GPR83* expression leads to earlier puberty timing.

### Joint regulation of ages at menarche and menopause

Previous GWASs have estimated a modest shared genetic etiology between AAM and age at natural menopause (ANM; genome-wide genetic correlation: $r_g = 0.14$; $P = 0.003$)[66]. ANM gene candidates are mainly expressed in the ovary and implicate DNA damage sensing and repair (DDR) processes that maintain genome stability and hence preserve the ovarian primordial follicle pool[67].

Of the 1,080 AAM signals, nine colocalized (at PP ≥ 0.5) and showed genome-wide significant ($P < 5 \times 10^{-8}$) association with ANM, and a further 11 AAM signals colocalized and showed association with ANM at $P < 4.6 \times 10^{-5}$ (= 0.05/1,080; Supplementary Table 32). We also considered whether ANM signals influence AAM. Of the 290 previously reported ANM signals[67], 21 colocalized and showed association with

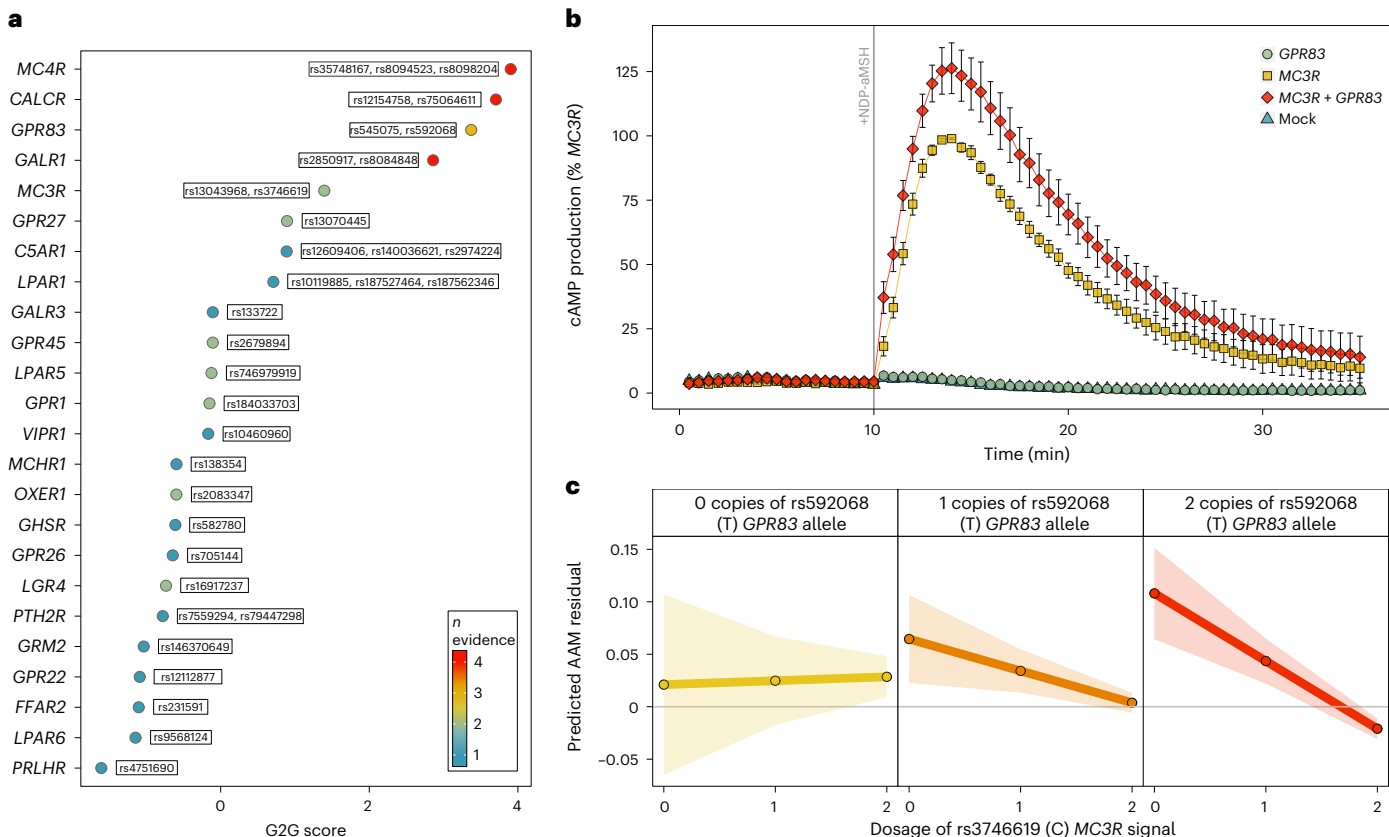

**Fig. 6 | Interactions between GPCRs on AAM. a**, A total of 24 brain-expressed GPCRs were implicated in AAM by the G2G analysis of the white European GWAS data. Point colors indicate the number of concordant G2G predictors implicating each GPCR. **b**, Time-resolved NDP-αMSH-stimulated cAMP production in HEK293 cells expressing *MC3R*-alone or with both *MC3R* and *GPR83*. Data are presented as the mean (±s.e.) percentage of the maximal *MC3R*-alone response (from six independent experiments). **c**, AAM according to dosage of *MC3R* function-increasing C alleles at rs3746619 (*x* axis in each panel) and *GPR83* expression-increasing T alleles at rs592068. Predicted means are represented by the lines, and the accompanying 95% CIs are denoted by the shaded areas. $\beta_{\text{interaction}} = -0.034 \pm 0.015$ years, $P_{\text{interaction}} = 0.02$. Supporting data are included in Supplementary Tables 28 and 30.

AAM at $P < 1.7 \times 10^{-4}$ (= 0.05/290), 13 of which were additional to the above AAM signals (Supplementary Table 33). Consistent with the phenotypic association between AAM and ANM[68], most of the shared signals (25/33) showed directionally concordant effects on AAM and ANM (shifting reproductive lifespan earlier or later). Several of these shared signals map to components of the HPG axis, including *GNRH1*, *INHBB* and *FSHB* (lead SNP rs11031006), which have previously reported associations with related reproductive phenotypes[8,69,70].

Several other shared AAM and ANM signals map to genes that encode components of DDR processes (*CHD4*, *CHEK2*, *DEPTOR*, *E2F1*, *MSH6*, *MSI2*, *PPARG*, *RAD18*, *RAD52*, *SCAI*, *SPRY4*, *SUMO1*, *TP53BP1*, *TRIP12* and *WWOX*; Supplementary Table 34), central to the establishment and maintenance of ovarian oocyte numbers[67], and not previously implicated in puberty timing. A notable example is rs199879971 ($P_{\text{AAM}} = 1.5 \times 10^{-20}$; $P_{\text{ANM}} = 1.5 \times 10^{-34}$), which is intronic in *MSH6*, a DNA mismatch repair gene that is primarily expressed in peripheral reproductive tissues, such as ovary and uterus (Supplementary Table 35). Furthermore, the colocalized ANM signal at *CHEK2* captures the previously described frameshift variant 1100delC[71]. This association was further supported by the exome data, with the 347 women carrying rare *CHEK2* PTVs (excluding 1100delC) reporting on average 2 months later AAM (s.e. = 0.99, $P = 0.04$). *CHEK2* encodes a cell cycle checkpoint inhibitor that has a crucial role in culling oocytes with unrepaired DNA damage[67].

A few of the shared AAM and ANM signals that map to DDR genes were assigned to the 'no early weight gain' trajectory (*CHD4*, *MSH6*, *SCAI* and *SUMO1*) and/or showed no association ($P > 0.05$) with adult BMI (*CHEK2*, *MSI2*, *PPARG* and *WWOX*; Supplementary Table 33). Three of

the shared AAM and ANM signals that map to DDR genes were assigned to the 'moderate early weight gain' trajectory and further colocalized with adult BMI (*RAD52*, *TP53BP1* and *TRIP12*; Supplementary Table 20). This suggests that some DDR genes might influence AAM via early life weight gain, although we only observed a trajectory-level DDR pathway enrichment for the 'no early weight gain' genes (Supplementary Tables 25 and 26).

## Discussion

The GWAS signals identified by this expanded multi-ancestry GWAS double the variance explained in AAM compared to previous findings[9] and highlight associations with consistent effects across the two studied ancestry groups. Furthermore, the common variant PGS contributes substantially to risks of extremely early and late puberty timing. Future studies should explore the potential of this PGS to predict extreme disorders of puberty timing, in contrast to the effects of known monogenic causes. While the majority of our sample was European, the inclusion of East Asian ancestry data increased our power to identify homogeneous signals across the two ancestries. Further studies, including individuals from a broader range of ancestry groups, will be required to understand how generalizable our findings are to non-European populations.

We describe the first systematic characterization of common genetic determinants of both ends of reproductive lifespan, AAM and ANM. The 33 identified shared signals highlight the concordant effects of HPG axis genes on both AAM and ANM and also the influence of ovary-expressed genes involved in DNA damage response.

DDR processes regulate ovarian oocyte numbers throughout life[67] but have not previously been implicated in puberty timing. DDR pathways were enriched among AAM genes that do not show a primary effect on early weight gain. Our findings suggest that the ovarian reserve, established during early fetal development, might signal centrally to influence the timing of puberty.

We address the considerable challenge of deriving biological insights from common variant signals[72] by developing G2G, an analytical pipeline that integrates a variety of data sources to enable gene prioritization. While comprehensive experimental validation is infeasible, its utility is supported by the prioritization of many genes with known involvement in sex hormone regulation and rare monogenic or syndromic disorders of puberty, obesity and hormone function. The validity of G2G prioritized genes is also supported by their enrichment for dynamic expression in GnRH neurons during their late stage of embryonic migration, when they begin their integration into the hypothalamic neural network controlling puberty[73]. Furthermore, we provide experimental support for one new high-scoring AAM gene, *GPR83*, which is co-expressed with, interacts with, and enhances the function of *MC3R*. Future studies should further explore the emerging role of brain-expressed GPCRs in linking central nutritional sensing to reproductive function.

Finally, we provide one of the few examples to date of epistatic interactions between common and rare genetic variants. Linked to puberty timing by both common and rare variants, the transcription factor *ZNF483* has diverse binding sites across the genome. We infer that greater *ZNF483* binding promotes earlier AAM, whereas rare deleterious variants in *ZNF483* appear to abolish the common genetic influence on puberty timing.

Together, these insights shed light on mechanisms, including early life weight gain and adiposity, hormone secretion and response and cellular susceptibility to DNA damage, that potentially explain the widely reported relationships between earlier puberty timing and higher risks of later life mortality, metabolic disease and cancer.

## Online content

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

Katherine A. Kentistou[1,212], Lena R. Kaisinger[1,212], Stasa Stankovic[1], Marc Vaudel[2,3], Edson Mendes de Oliveira[4], Andrea Messina[5,6], Robin G. Walters[7], Xiaoxi Liu[8], Alexander S. Busch[9,10], Hannes Helgason[11,12], Deborah J. Thompson[13], Federico Santoni[5,6], Konstantin M. Petricek[14], Yassine Zouaghi[5,6], Isabel Huang-Doran[4], Daniel F. Gudbjartsson[11,12], Eirik Bratland[2,15], Kuang Lin[7], Eugene J. Gardner[1], Yajie Zhao[1], Raina Y. Jia[1], Chikashi Terao[8,16,17], Marjorie J. Riggan[18], Manjeet K. Bolla[13], Mojgan Yazdanpanah[19], Nahid Yazdanpanah[19], Jonathan P. Bradfield[20,21], Linda Broer[22,23], Archie Campbell[24], Daniel I. Chasman[25], Diana L. Cousminer[26,27,28], Nora Franceschini[29], Lude H. Franke[30], Giorgia Girotto[31,32], Chunyan He[33,34], Marjo-Riitta Järvelin[35,36,37,38,39], Peter K. Joshi[40], Yoichiro Kamatani[41], Robert Karlsson[42], Jian'an Luan[1], Kathryn L. Lunetta[43,44], Reedik Mägi[45], Massimo Mangino[46,47], Sarah E. Medland[48,49,50], Christa Meisinger[51], Raymond Noordam[52], Teresa Nutile[53], Maria Pina Concas[31], Ozren Polašek[54,55], Eleonora Porcu[56,57], Susan M. Ring[58,59], Cinzia Sala[60], Albert V. Smith[61,62], Toshiko Tanaka[63], Peter J. van der Most[64], Veronique Vitart[65], Carol A. Wang[66,67], Gonneke Willemsen[68], Marek Zygmunt[69], Thomas U. Ahearn[70], Irene L. Andrulis[71,72], Hoda Anton-Culver[73], Antonis C. Antoniou[13], Paul L. Auer[74], Catriona L. K. Barnes[40], Matthias W. Beckmann[75], Amy Berrington de Gonzalez[76], Natalia V. Bogdanova[77,78,79], Stig E. Bojesen[80,81], Hermann Brenner[82,83,84], Julie E. Buring[25], Federico Canzian[85], Jenny Chang-Claude[86,87], Fergus J. Couch[88], Angela Cox[89], Laura Crisponi[56], Kamila Czene[42], Mary B. Daly[90], Ellen W. Demerath[91], Joe Dennis[13], Peter Devilee[92,93], Immaculata De Vivo[94,95], Thilo Dörk[78], Alison M. Dunning[96], Miriam Dwek[97], Johan G. Eriksson[98,99,100], Peter A. Fasching[75], Lindsay Fernandez-Rhodes[101], Liana Ferreli[56], Olivia Fletcher[102], Manuela Gago-Dominguez[103], Montserrat García-Closas[70], José A. García-Sáenz[104], Anna González-Neira[105], Harald Grallert[106,107], Pascal Guénel[108], Christopher A. Haiman[109], Per Hall[42,110], Ute Hamann[111], Hakon Hakonarson[21,26,112,113], Roger J. Hart[114], Martha Hickey[115], Maartje J. Hooning[116], Reiner Hoppe[117,118], John L. Hopper[119], Jouke-Jan Hottenga[68], Frank B. Hu[95,120], Hanna Huebner[75], David J. Hunter[7,94], ABCTB Investigators[121,*], Helena Jernström[122], Esther M. John[123,124], David Karasik[125,126], Elza K. Khusnutdinova[127,128], Vessela N. Kristensen[129,130], James V. Lacey[131,132], Diether Lambrechts[133,134], Lenore J. Launer[135], Penelope A. Lind[48,50,136], Annika Lindblom[137,138], Patrik K. E. Magnusson[42], Arto Mannermaa[139,140], Mark I. McCarthy[141,142,143], Thomas Meitinger[144], Cristina Menni[46], Kyriaki Michailidou[13,145], Iona Y. Millwood[7], Roger L. Milne[119,146], Grant W. Montgomery[147], Heli Nevanlinna[148], Ilja M. Nolte[64], Dale R. Nyholt[149], Nadia Obi[150,151], Katie M. O'Brien[152], Kenneth Offit[153,154], Albertine J. Oldehinkel[155], Sisse R. Ostrowski[156,157], Aarno Palotie[158,159,160,161,162,163], Ole B. Pedersen[157,164], Annette Peters[107,165], Giulia Pianigiani[31], Dijana Plaseska-Karanfilska[166], Anneli Pouta[167], Alfred Pozarickij[7], Paolo Radice[168], Gad Rennert[169], Frits R. Rosendaal[170], Daniela Ruggiero[53,171], Emmanouil Saloustros[172], Dale P. Sandler[152], Sabine Schipf[173], Carsten O. Schmidt[173], Marjanka K. Schmidt[174,175], Kerrin Small[46], Beatrice Spedicati[32], Meir Stampfer[94,95], Jennifer Stone[119,176], Rulla M. Tamimi[94,177], Lauren R. Teras[178], Emmi Tikkanen[163,179], Constance Turman[94,180], Celine M. Vachon[181], Qin Wang[13], Robert Winqvist[182,183], Alicja Wolk[184], Babette S. Zemel[112,185], Wei Zheng[186], Ko W. van Dijk[93,187], Behrooz Z. Alizadeh[64], Stefania Bandinelli[188], Eric Boerwinkle[189], Dorret I. Boomsma[68,190], Marina Ciullo[53,171], Georgia Chenevix-Trench[48], Francesco Cucca[56,57], Tõnu Esko[45], Christian Gieger[106,107,191], Struan F. A. Grant[26,27,28,112,192], Vilmundur Gudnason[61,62], Caroline Hayward[65], Ivana Kolčić[54,55], Peter Kraft[94,180], Deborah A. Lawlor[58,59], Nicholas G. Martin[48], Ellen A. Nøhr[193], Nancy L. Pedersen[42], Craig E. Pennell[66,67,194], Paul M. Ridker[25], Antonietta Robino[31], Harold Snieder[64], Ulla Sovio[35,195], Tim D. Spector[46], Doris Stöckl[107,196], Cathie Sudlow[24,197], Nic J. Timpson[58,59], Daniela Toniolo[60], André Uitterlinden[22,23], Sheila Ulivi[31], Henry Völzke[173], Nicholas J. Wareham[1], Elisabeth Widen[163], James F. Wilson[40], The Lifelines Cohort Study*, The Danish Blood Donor Study*, The Ovarian Cancer Association Consortium*, The Breast Cancer Association Consortium*, The Biobank Japan Project*, The China Kadoorie Biobank Collaborative Group*, Paul D. P. Pharoah[13,96], Liming Li[198,199], Douglas F. Easton[13,96], Pål R. Njølstad[2,200], Patrick Sulem[11], Joanne M. Murabito[44,201], Anna Murray[202], Despoina Manousaki[203,204,205], Anders Juul[10,206,207], Christian Erikstrup[208,209], Kari Stefansson[11,62], Momoko Horikoshi[210], Zhengming Chen[7], I. Sadaf Farooqi[4], Nelly Pitteloud[5,6], Stefan Johansson[2,15], Felix R. Day[1,212], John R. B. Perry[1,211,212]✉ & Ken K. Ong[1,212,212]

[1]MRC Epidemiology Unit, University of Cambridge School of Clinical Medicine, Institute of Metabolic Science, Cambridge Biomedical Campus, Cambridge, UK. [2]Mohn Center for Diabetes Precision Medicine, Department of Clinical Science, University of Bergen, Bergen, Norway. [3]Department of Genetics and Bioinformatics, Health Data and Digitalization, Norwegian Institute of Public Health, Oslo, Norway. [4]University of Cambridge Metabolic Research Laboratories and NIHR Cambridge Biomedical Research Centre, Wellcome-MRC Institute of Metabolic Science, Addenbrooke's Hospital, Cambridge, UK. [5]Division of Endocrinology, Diabetology, and Metabolism, Lausanne University Hospital, Lausanne, Switzerland. [6]Faculty of Biology and Medicine, University of Lausanne, Lausanne, Switzerland. [7]Nuffield Department of Population Health, University of Oxford, Oxford, UK. [8]Laboratory for Statistical and Translational Genetics, RIKEN Center for Integrative Medical Sciences, Yokohama, Japan. [9]Department of General Pediatrics, University of Münster, Münster, Germany. [10]Department of Growth and Reproduction, Copenhagen University Hospital—Rigshospitalet, Copenhagen, Denmark. [11]deCODE Genetics/Amgen, Inc., Reykjavik, Iceland. [12]School of Engineering and Natural Sciences, University of Iceland, Reykjavik, Iceland. [13]Centre for Cancer Genetic Epidemiology, Department of Public Health and Primary Care, University of Cambridge, Cambridge, UK. [14]Charité Universitätsmedizin Berlin, Corporate Member of Freie Universität Berlin and Humboldt-Universität zu Berlin, Institute of Pharmacology, Berlin, Germany. [15]Department of Medical Genetics, Haukeland University Hospital, Bergen, Norway. [16]Clinical Research Center, Shizuoka General Hospital, Shizuoka, Japan. [17]Department of Applied Genetics, The School of Pharmaceutical Sciences, University of Shizuoka, Shizuoka, Japan. [18]Department of Gynecology, Duke University Medical Center, Durham, NC, USA. [19]Research Center of the Sainte-Justine University Hospital, University of Montreal, Montreal, Quebec, Canada. [20]Quantinuum Research, Wayne, PA, USA. [21]Center for Applied Genomics, Children's Hospital of Philadelphia, Philadelphia, PA, USA. [22]Department of Internal Medicine, Erasmus MC, Rotterdam, The Netherlands. [23]Department of Epidemiology, Erasmus MC, Rotterdam, The Netherlands. [24]Centre for Genomic and Experimental Medicine, Institute of Genetics and Cancer, University of Edinburgh, Edinburgh, UK. [25]Division of Preventive Medicine, Brigham and Women's Hospital and Harvard Medical School, Boston, MA, USA. [26]Division of Human Genetics, Children's Hospital of Philadelphia, Philadelphia, PA, USA. [27]Department of Genetics, University of Pennsylvania, Philadelphia, PA, USA. [28]Center for Spatial and Functional Genomics, Children's Hospital of Philadelphia, Philadelphia, PA, USA. [29]Department of Epidemiology, University of North Carolina, Chapel Hill, NC, USA. [30]Department of Genetics, University of Groningen, University Medical Center Groningen, Groningen, The Netherlands. [31]Institute for Maternal and Child Health—IRCCS 'Burlo Garofolo', Trieste, Italy. [32]Department of Medicine, Surgery and Health Sciences, University of Trieste, Trieste, Italy. [33]Department of Internal Medicine, Division of Medical Oncology, University of Kentucky College of Medicine, Lexington, KY, USA. [34]Cancer Prevention and Control Research Program, Markey Cancer Center, University of Kentucky, Lexington, KY, USA. [35]Department of Epidemiology and Biostatistics, MRC Health Protection Agency (HPA) Centre for Environment and Health, School of Public Health, Imperial College London, London, UK. [36]Institute of Health Sciences, University of Oulu, Oulu, Finland. [37]Biocenter Oulu, University of Oulu, Oulu, Finland. [38]Unit of Primary Care, Oulu University Hospital, Oulu, Finland. [39]Department of Children and Young People and Families, National Institute for Health and Welfare, Oulu, Finland. [40]Centre for Global Health Research, Usher Institute, University of Edinburgh, Edinburgh, Scotland. [41]Laboratory of Complex Trait Genomics, Department of Computational Biology and Medical Sciences, Graduate School of Frontier Sciences, The University of Tokyo, Tokyo, Japan. [42]Department of Medical Epidemiology and Biostatistics, Karolinska Institutet, Stockholm, Sweden. [43]Department of Biostatistics, Boston University School of Public Health, Boston, MA, USA. [44]NHLBI's and Boston University's Framingham Heart Study, Framingham, MA, USA. [45]Institute of Genomics, University of Tartu, Tartu, Estonia. [46]Department of Twin Research and Genetic Epidemiology, King's College London, London, UK. [47]NIHR Biomedical Research Centre at Guy's and St. Thomas' Foundation Trust, London, UK. [48]QIMR Berghofer Medical Research Institute, Brisbane, Queensland, Australia. [49]School of Psychology, University of Queensland, Brisbane, Queensland, Australia. [50]Faculty of Medicine, University of Queensland, Brisbane, Queensland, Australia. [51]Epidemiology, Medical Faculty, University of Augsburg, University Hospital of Augsburg, Augsburg, Germany. [52]Department of Internal Medicine, Section of Gerontology and Geriatrics, Leiden University Medical Center, Leiden, The Netherlands. [53]Institute of Genetics and Biophysics 'A. Buzzati-Traverso', CNR, Naples, Italy. [54]University of Split School of Medicine, Split, Croatia. [55]Algebra University College, Zagreb, Croatia. [56]Institute of Genetics and Biomedical Research, National Research Council, Sardinia, Italy. [57]Department of Biomedical Sciences, University of Sassari, Sassari, Italy. [58]MRC Integrative Epidemiology Unit, University of Bristol, Bristol, UK. [59]Population Health Science, Bristol Medical School, University of Bristol, Bristol, UK. [60]Division of Genetics and Cell Biology, San Raffaele Hospital, Milano, Italy. [61]Icelandic Heart Association, Kopavogur, Iceland. [62]Faculty of Medicine, University of Iceland, Reykjavik, Iceland. [63]National Institute on Aging, National Institutes of Health, Baltimore, MD, USA. [64]Department of Epidemiology, University of Groningen, University Medical Center Groningen, Groningen, The Netherlands. [65]MRC Human Genetics Unit, Institute of Genetics and Cancer, University of Edinburgh, Edinburgh, UK. [66]School of Medicine and Public Health, University of Newcastle, Newcastle, New South Wales, Australia. [67]Hunter Medical Research Institute, Newcastle, New South Wales, Australia. [68]Department of Biological Psychology, Vrije Universiteit Amsterdam; Amsterdam Public Health (APH) Research Institute, Amsterdam, The Netherlands. [69]Clinic of Gynaecology and Obstetrics, University Medicine Greifswald, Greifswald, Germany. [70]Division of Cancer Epidemiology and Genetics National Cancer Institute, National Institutes of Health, Department of Health and Human Services Bethesda, Bethesda, MD, USA. [71]Fred A. Litwin Center for Cancer Genetics, Lunenfeld-Tanenbaum Research Institute of Mount Sinai Hospital, Toronto, Ontario, Canada. [72]Department of Molecular Genetics, University of Toronto, Toronto, Ontario, Canada. [73]Department of Medicine, Genetic Epidemiology Research Institute, University of California Irvine, Irvine, CA, USA. [74]Division of Biostatistics, Institute for Health and Equity and Cancer Center, Medical College of Wisconsin, Milwaukee, WI, USA. [75]Department of Gynecology and Obstetrics, Comprehensive Cancer Center Erlangen-EMN, Friedrich-Alexander University Erlangen-Nuremberg, University Hospital Erlangen, Erlangen, Germany. [76]Division of Genetics and Epidemiology, The Institute of Cancer Research, London, UK. [77]Department of Radiation Oncology, Hannover Medical School, Hannover, Germany. [78]Gynaecology Research Unit, Hannover Medical School, Hannover, Germany. [79]N.N. Alexandrov Research Institute of Oncology and Medical Radiology, Minsk, Belarus. [80]Copenhagen General Population Study, Herlev and Gentofte Hospital Copenhagen University Hospital, Herlev, Denmark. [81]Department of Clinical Biochemistry, Herlev and Gentofte Hospital Copenhagen University Hospital, Herlev, Denmark. [82]Division of Clinical Epidemiology and Aging Research, German Cancer Research Center (DKFZ), Heidelberg, Germany. [83]Division of Preventive Oncology, German Cancer Research Center (DKFZ) and National Center for Tumor Diseases (NCT), Heidelberg, Germany. [84]German Cancer Consortium (DKTK), German Cancer Research Center (DKFZ), Heidelberg, Germany. [85]Genomic Epidemiology Group, German Cancer Research Center (DKFZ), Heidelberg, Germany. [86]Division of Cancer Epidemiology, German Cancer Research Center (DKFZ), Heidelberg, Germany. [87]Cancer Epidemiology Group, University Cancer Center Hamburg (UCCH), University Medical Center Hamburg-Eppendorf, Hamburg, Germany. [88]Department of Laboratory Medicine and Pathology, Mayo Clinic Rochester, Rochester, MN, USA. [89]Sheffield Institute for Nucleic Acids (SInFoNiA), Department of Oncology and Metabolism, University of Sheffield, Sheffield, UK. [90]Department of Clinical Genetics, Fox Chase Cancer Center, Philadelphia, PA, USA. [91]Division of Epidemiology and Community Health, School of Public Health, University of Minnesota, Minneapolis, MN, USA. [92]Department of Pathology, Leiden University Medical Center, Leiden, The Netherlands. [93]Department of Human Genetics, Leiden University Medical Center, Leiden, The Netherlands. [94]Department of Epidemiology, Harvard T.H. Chan School of Public Health, Boston, MA, USA. [95]Channing Division of Network Medicine, Department of Medicine, Brigham and Women's Hospital and Harvard Medical School, Boston, MA, USA.

[96]Centre for Cancer Genetic Epidemiology, Department of Oncology, University of Cambridge, Cambridge, UK. [97]School of Life Sciences, University of Westminster, London, UK. [98]Department of General Practice and Primary Healthcare, University of Helsinki, Helsinki University Hospital, Helsinki, Finland. [99]Yong Loo Lin School of Medicine, Department of Obstetrics and Gynecology and Human Potential Translational Research Programme, National University Singapore, Singapore City, Singapore. [100]Singapore Institute for Clinical Sciences (SICS), Agency for Science, Technology and Research (A*STAR), Singapore City, Singapore. [101]Department of Biobehavioral Health, Pennsylvania State University, University Park, PA, USA. [102]The Breast Cancer Now Toby Robins Research Centre, The Institute of Cancer Research, London, UK. [103]Genomic Medicine Group, International Cancer Genetics and Epidemiology Group Fundación Pública Galega de Medicina Xenómica, Instituto de Investigación Sanitaria de Santiago de Compostela (IDIS), Complejo Hospitalario Universitario de Santiago, SERGAS Santiago de Compostela, Coruña, Spain. [104]Medical Oncology Department, Hospital Clínico San Carlos Instituto de Investigación Sanitaria San Carlos (IdISSC), Centro Investigación Biomédica en Red de Cáncer (CIBERONC), Madrid, Spain. [105]Human Genotyping Unit-CeGen, Spanish National Cancer Research Centre (CNIO), Madrid, Spain. [106]Research Unit of Molecular Epidemiology, Helmholtz Zentrum München—German Research Center for Environmental Health, Neuherberg, Germany. [107]Institute of Epidemiology, Helmholtz Zentrum München—German Research Center for Environmental Health, Neuherberg, Germany. [108]Team 'Exposome and Heredity', CESP, Gustave Roussy INSERM, University Paris-Saclay, UVSQ, Orsay, France. [109]Department of Preventive Medicine, Keck School of Medicine, University of Southern California, Los Angeles, CA, USA. [110]Department of Oncology, Södersjukhuset, Stockholm, Sweden. [111]Molecular Genetics of Breast Cancer, German Cancer Research Center (DKFZ), Heidelberg, Germany. [112]Department of Pediatrics, University of Pennsylvania Perelman School of Medicine, Philadelphia, PA, USA. [113]Division of Pulmonary Medicine, Children's Hospital of Philadelphia, Philadelphia, PA, USA. [114]Division of Obstetrics and Gynaecology, University of Western Australia, Crawley, Western Australia, Australia. [115]Department of Obstetrics and Gynaecology, University of Melbourne and The Royal Women's Hospital, Parkville, Victoria, Australia. [116]Department of Medical Oncology, Erasmus MC Cancer Institute, Rotterdam, The Netherlands. [117]Dr. Margarete Fischer-Bosch-Institute of Clinical Pharmacology, Stuttgart, Germany. [118]University of Tübingen, Tübingen, Germany. [119]Centre for Epidemiology and Biostatistics, Melbourne School of Population and Global Health, The University of Melbourne, Melbourne, Victoria, Australia. [120]Department of Nutrition, Harvard T.H. Chan School of Public Health School of Public Health, Boston, MA, USA. [121]Australian Breast Cancer Tissue Bank, Westmead Institute for Medical Research, University of Sydney, Sydney, New South Wales, Australia. [122]Oncology, Department of Clinical Sciences in Lund, Lund University, Lund, Sweden. [123]Department of Epidemiology and Population Health, Stanford University School of Medicine Stanford, Stanford, CA, USA. [124]Department of Medicine, Division of Oncology Stanford Cancer Institute, Stanford University School of Medicine Stanford, Stanford, CA, USA. [125]Hebrew SeniorLife Institute for Aging Research, Boston, MA, USA. [126]Harvard Medical School, Boston, MA, USA. [127]Institute of Biochemistry and Genetics of the Ufa Federal Research Centre of the Russian Academy of Sciences, Ufa, Russia. [128]Department of Genetics and Fundamental Medicine, Bashkir State University, Ufa, Russia. [129]Department of Medical Genetics, Oslo University Hospital and University of Oslo, Oslo, Norway. [130]Institute of Clinical Medicine, Faculty of Medicine, University of Oslo, Oslo, Norway. [131]Department of Computational and Quantitative Medicine, City of Hope, Duarte, CA, USA. [132]City of Hope Comprehensive Cancer Center, City of Hope, Duarte, CA, USA. [133]Laboratory for Translational Genetics, Department of Human Genetics, KU Leuven, Leuven, Belgium. [134]VIB Center for Cancer Biology, VIB, Leuven, Belgium. [135]Laboratory of Epidemiology and Population Sciences, National Institute on Aging, Intramural Research Program, National Institutes of Health, Bethesda, MD, USA. [136]School of Biomedical Sciences, Queensland University of Technology, Brisbane, Queensland, Australia. [137]Department of Molecular Medicine and Surgery, Karolinska Institutet, Stockholm, Sweden. [138]Department of Clinical Genetics, Karolinska University Hospital, Stockholm, Sweden. [139]Translational Cancer Research Area, University of Eastern Finland, Kuopio, Finland. [140]Institute of Clinical Medicine, Pathology and Forensic Medicine, University of Eastern Finland, Kuopio, Finland. [141]Wellcome Trust Centre for Human Genetics, University of Oxford, Oxford, UK. [142]Oxford Centre for Diabetes, Endocrinology and Metabolism, University of Oxford, Churchill Hospital, Oxford, UK. [143]NIHR Oxford Biomedical Research Centre, Churchill Hospital, Oxford, UK. [144]Institute of Human Genetics, Klinikum rechts der Isar, Technical University of Munich, School of Medicine, Munich, Germany. [145]Biostatistics Unit, The Cyprus Institute of Neurology and Genetics, Nicosia, Cyprus. [146]Cancer Epidemiology Division, Cancer Council Victoria, Melbourne, Victoria, Australia. [147]Institute for Molecular Bioscience, The University of Queensland, Brisbane, Queensland, Australia. [148]Department of Obstetrics and Gynecology, Helsinki University Hospital, University of Helsinki, Helsinki, Finland. [149]School of Biomedical Sciences, Faculty of Health, Centre for Genomics and Personalised Health, Queensland University of Technology, Brisbane, Queensland, Australia. [150]Institute for Occupational Medicine and Maritime Medicine, University Medical Center Hamburg-Eppendorf, Hamburg, Germany. [151]Institute of Medical Biometry and Epidemiology, University Medical Center Hamburg-Eppendorf, Hamburg, Germany. [152]Epidemiology Branch, National Institute of Environmental Health Sciences, NIH Research Triangle Park, Durham, NC, USA. [153]Clinical Genetics Research Lab, Department of Cancer Biology and Genetics, Memorial Sloan Kettering Cancer Center, New York City, NY, USA. [154]Clinical Genetics Service, Department of Medicine, Memorial Sloan Kettering Cancer Center, New York City, NY, USA. [155]Interdisciplinary Center Psychopathology and Emotion Regulation, University Medical Center Groningen, University of Groningen, Groningen, The Netherlands. [156]Department of Clinical Immunology, Rigshospitalet—University of Copenhagen, Copenhagen, Denmark. [157]Department of Clinical Medicine, Faculty of Health and Medical Sciences, University of Copenhagen, Copenhagen, Denmark. [158]Psychiatric and Neurodevelopmental Genetics Unit, Massachusetts General Hospital and Harvard Medical School, Boston, MA, USA. [159]Medical and Population Genetics Program, Broad Institute of MIT and Harvard, Cambridge, MA, USA. [160]Stanley Center for Psychiatric Research, Broad Institute of MIT and Harvard, Cambridge, MA, USA. [161]Wellcome Trust Sanger Institute, Wellcome Trust Genome Campus, Hinxton, UK. [162]Analytic and Translational Genetics Unit, Massachusetts General Hospital and Harvard Medical School, Boston, MA, USA. [163]Institute for Molecular Medicine Finland (FIMM), University of Helsinki, Helsinki, Finland. [164]Department of Clinical Immunology, Zealand University Hospital, Køge, Denmark. [165]Institute for Medical Information Processing, Biometry and Epidemiology—IBE, Ludwig-Maximilians-Universität München, Munich, Germany. [166]Research Centre for Genetic Engineering and Biotechnology 'Georgi D. Efremov', MASA, Skopje, Republic of North Macedonia. [167]National Institute for Health and Welfare, Helsinki, Finland. [168]Unit of Preventive Medicine: Molecular Bases of Genetic Risk, Department of Experimental Oncology, Fondazione IRCCS, Istituto Nazionale dei Tumori (INT), Milan, Italy. [169]Faculty of Medicine, Clalit National Cancer Control Center, Carmel Medical Center and Technion, Haifa, Israel. [170]Department of Clinical Epidemiology, Leiden University Medical Center, Leiden, The Netherlands. [171]IRCCS Neuromed, Isernia, Italy. [172]Division of Oncology, Faculty of Medicine, School of Health Sciences, University of Thessaly, Larissa, Greece. [173]Institute for Community Medicine, University Medicine Greifswald, Greifswald, Germany. [174]Division of Molecular Pathology, The Netherlands Cancer Institute, Amsterdam, The Netherlands. [175]Division of Psychosocial Research and Epidemiology, The Netherlands Cancer Institute—Antoni van Leeuwenhoek Hospital, Amsterdam, The Netherlands. [176]Genetic Epidemiology Group, School of Population and Global Health, University of Western Australia Perth, Perth, Western Australia, Australia. [177]Department of Population Health Sciences, Weill Cornell Medicine, New York City, NY, USA. [178]Department of Population Science, American Cancer Society, Atlanta, GA, USA. [179]Public Health Genomics Unit, Department of Chronic Disease Prevention, National Institute for Health and Welfare, Helsinki, Finland. [180]Program in Genetic Epidemiology and Statistical Genetics, Harvard T.H. Chan School of Public Health, Boston, MA, USA. [181]Department of Quantitative Health Sciences, Division of Epidemiology, Mayo Clinic Rochester, Rochester, MN, USA.

[182]Laboratory of Cancer Genetics and Tumor Biology, Translational Medicine Research Unit, Biocenter Oulu, University of Oulu, Oulu, Finland. [183]Laboratory of Cancer Genetics and Tumor Biology, Northern Finland Laboratory Centre, Oulu, Finland. [184]Institute of Environmental Medicine, Karolinska Institutet, Stockholm, Sweden. [185]Division of Gastroenterology, Hepatology and Nutrition, Children's Hospital of Philadelphia, Philadelphia, PA, USA. [186]Division of Epidemiology, Department of Medicine, Vanderbilt Epidemiology Center, Vanderbilt-Ingram Cancer Center, Vanderbilt University School of Medicine, Nashville, TN, USA. [187]Department of Internal Medicine, Division of Endocrinology, Leiden University Medical Center, Leiden, The Netherlands. [188]Geriatric Unit, Local Health Toscana Centro, Florence, Italy. [189]Human Genetics Center, University of Texas Health Science Center at Houston, Houston, TX, USA. [190]Amsterdam Reproduction and Development Research Institute, Amsterdam, The Netherlands. [191]German Center for Diabetes Research (DZD), Neuherberg, Germany. [192]Division of Endocrinology and Diabetes, Children's Hospital of Philadelphia, Philadelphia, PA, USA. [193]Institute of Clinical Research, University of Southern Denmark, Department of Obstetrics and Gynecology, Odense University Hospital, Odense, Denmark. [194]Department of Maternity and Gynaecology, John Hunter Hospital, Newcastle, New South Wales, Australia. [195]Department of Obstetrics and Gynaecology, University of Cambridge, Cambridge, UK. [196]State Institute of Health, Bavarian Health and Food Safety Authority (LGL), Oberschleissheim, Germany. [197]Centre for Medical Informatics, Usher Institute, University of Edinburgh, Edinburgh, UK. [198]Department of Epidemiology and Biostatistics, School of Public Health, Peking University, Beijing, China. [199]Center for Public Health and Epidemic Preparedness and Response, Peking University, Beijing, China. [200]Children and Adolescent Clinic, Haukeland University Hospital, Bergen, Norway. [201]Boston University Chobanian and Avedisian School of Medicine, Department of Medicine, Section of General Internal Medicine, Boston, MA, USA. [202]Genetics of Complex Traits, University of Exeter Medical School, University of Exeter, RILD Level 3, Royal Devon and Exeter Hospital, Exeter, UK. [203]Centre Hospitalier Universitaire (CHU) Sainte-Justine Research Center, University of Montreal, Montreal, Quebec, Canada. [204]Department of Pediatrics, University of Montreal, Montreal, Quebec, Canada. [205]Department of Biochemistry and Molecular Medicine, University of Montreal, Montreal, Quebec, Canada. [206]International Center for Research and Research Training in Endocrine Disruption of Male Reproduction and Child Health (EDMaRC), Copenhagen University Hospital—Rigshospitalet, Copenhagen, Denmark. [207]Department of Clinical Medicine, University of Copenhagen, Copenhagen, Denmark. [208]Department of Clinical Immunology, Aarhus University Hospital, Aarhus, Denmark. [209]Department of Clinical Medicine, Aarhus University, Aarhus, Denmark. [210]Laboratory for Genomics of Diabetes and Metabolism, RIKEN Center for Integrative Medical Sciences, Yokohama, Japan. [211]Metabolic Research Laboratory, Wellcome-MRC Institute of Metabolic Science, University of Cambridge School of Clinical Medicine, Cambridge, UK. [212]Department of Paediatrics, University of Cambridge, Cambridge, UK. [212]These authors contributed equally: Katherine A. Kentistou, Lena R. Kaisinger, Felix R. Day, John R. B. Perry, Ken K. Ong. *Full lists of authors and their affiliations appear at the end of the paper. ✉e-mail: john.perry@mrc-epid.cam.ac.uk

## ABCTB Investigators

**Manjeet K. Bolla[13]**

Full lists of members and their affiliations appear in the Supplementary Information.

## The Lifelines Cohort Study

**Behrooz Z. Alizadeh[64], Lude H. Franke[30] & Harold Snieder[64]**

Full lists of members and their affiliations appear in the Supplementary Information.

## The Danish Blood Donor Study

**Christian Erikstrup[208,209], Daniel F. Gudbjartsson[11,12], Sisse R. Ostrowski[156,157], Ole B. Pedersen[157,164] & Kari Stefansson[11,62]**

Full lists of members and their affiliations appear in the Supplementary Information.

## The Ovarian Cancer Association Consortium

**Irene L. Andrulis[71,72], Hoda Anton-Culver[73], Antonis C. Antoniou[13], Matthias W. Beckmann[75], Natalia V. Bogdanova[77,78,79], Jenny Chang-Claude[86,87], Georgia Chenevix-Trench[48], Fergus J. Couch[88], Mary B. Daly[90], Joe Dennis[13], Thilo Dörk[78], Douglas F. Easton[13,96], Peter A. Fasching[75], Christopher A. Haiman[109], Ute Hamann[111], Maartje J. Hooning[116], David J. Hunter[7,94], Helena Jernström[122], Esther M. John[123], Elza K. Khusnutdinova[127,128], Diether Lambrechts[133,134], Roger L. Milne[119,146], Heli Nevanlinna[148], Kenneth Offit[153,154], Paul D. P. Pharoah[13,96], Paolo Radice[168], Gad Rennert[169], Marjorie J. Riggan[18], Dale P. Sandler[152] & Alicja Wolk[184]**

Full lists of members and their affiliations appear in the Supplementary Information.

## The Breast Cancer Association Consortium

**Irene L. Andrulis[71,72], Hoda Anton-Culver[73], Antonis C. Antoniou[13], Paul L. Auer[74], Stig E. Bojesen[80,81], Manjeet K. Bolla[13], Hermann Brenner[82,83,84], Federico Canzian[85], Jenny Chang-Claude[86,87], Georgia Chenevix-Trench[48], Fergus J. Couch[88], Angela Cox[89], Kamila Czene[42], Mary B. Daly[90], Joe Dennis[13], Peter Devilee[92,93], Thilo Dörk[78], Alison M. Dunning[96], Miriam Dwek[97], Douglas F. Easton[13,96], Peter A. Fasching[75], Manuela Gago-Dominguez[103], Montserrat García-Closas[70], José A. García-Sáenz[104], Pascal Guénel[108], Christopher A. Haiman[109], Per Hall[110], Ute Hamann[111], Maartje J. Hooning[116], Reiner Hoppe[117,118], John L. Hopper[119], David J. Hunter[7,94], Helena Jernström[122], Esther M. John[123], Peter Kraft[94,180], Vessela N. Kristensen[129,130], Diether Lambrechts[133,134], Annika Lindblom[137,138], Arto Mannermaa[139,140], Kyriaki Michailidou[13,145], Roger L. Milne[119,146], Heli Nevanlinna[148],**

Paul D. P. Pharoah[13,96], Dijana Plaseska-Karanfilska[166], Paolo Radice[168], Gad Rennert[169], Emmanouil Saloustros[172], Dale P. Sandler[152], Marjanka K. Schmidt[174,175], Jennifer Stone[119,176], Rulla M. Tamimi[94,177], Celine M. Vachon[181], Qin Wang[13], Robert Winqvis[182,183], Alicja Wolk[184] & Wei Zheng[186]

Full lists of members and their affiliations appear in the Supplementary Information.

## The Biobank Japan Project

Momoko Horikoshi[210], Yoichiro Kamatani[41], Xiaoxi Liu[8] & Chikashi Terao[8,16,17]

Full lists of members and their affiliations appear in the Supplementary Information.

## The China Kadoorie Biobank Collaborative Group

Zhengming Chen[7], Liming Li[198,199], Kuang Lin[7] & Robin G. Walters[7]

Full lists of members and their affiliations appear in the Supplementary Information.

## Methods

UK Biobank data have approval from the North West Multicentre Research Ethics Committee as a Research Tissue Bank. The 23andMe research participants provided informed consent and volunteered to participate in the research online under a protocol approved by the external Association for the Accreditation of Human Research Protection Programs (AAHRPP) accredited Institutional Review Board (IRB), Ethical and Independent Review Services. Each of the other individual studies that contributed data has its own ethical approval from the relevant boards.

### GWAS meta-analysis for AAM

Association summary statistics were collated from studies on AAM (predominantly recalled in adulthood) and genome-wide SNP arrays imputed to the 1000 Genomes reference panel or more recent (Supplementary Table 1). Genetic variants and individuals were filtered based on study-specific quality control metrics. In each study, genetic variants were tested for association with AAM in additive linear regression models, including as covariates age and any study-specific variables, such as genetic principal components. Insertion and deletion polymorphisms were coded as 'I' and 'D' to allow harmonization across all studies. Association statistics for each SNP were then processed centrally using a standardized quality control pipeline[74]. Each variant was meta-analyzed using a fixed-effects inverse-variance-weighted (IVW) model using METAL[75]. This was done in two stages. First, summary statistics from studies within each stratum ((1) ReproGen consortium studies, (2) reproductive cancer consortium studies and (3) East Asian studies) were meta-analyzed and then filtered to include only variants present in more than half of the studies within each stratum. Second, strata-level results were meta-analyzed with data from UK Biobank[76], using 'first instance' data for AAM (field 2714) and 23andMe. Initially, we performed a European-only analysis (up to $n = 632{,}955$). This combined file was filtered to include only variants present in the UK Biobank and at least one other stratum. Variants were also filtered to include MAF $\geq 0.1\%$. We then performed a second analysis by adding the data from the East Asian studies (up to $n = 799{,}845$) and followed the same sample filtering steps and identification of independent signals (described below).

### Replication and explained variance

Independent replication of identified signals was performed in data from the DBDS[16]. The DBDS includes questionnaire-recalled AAM data on 35,472 European women (Age when menstruation started?). Mean age at recall was 38.4 years (s.d. = 12.9 years), and the mean AAM was 13.1 years (s.d. = 1.4 years). Indirect confirmation of AAM signals was sought by association with AVB in men in UK Biobank[17] ($n = 191{,}235$ European men – data field 2385), the 23andMe study[18] ($n = 55{,}781$ European men) and a meta-analysis of the two[14] ($n = 205{,}354$ European men). For signals with missing data for either AVB dataset, we identified proxies using the UK Biobank White European dataset (within 1 Mb of the reported signal and $r^2 > 0.6$), choosing the variant with the highest $r^2$ value. Given the smaller sample sizes of these cohorts, we performed a binomial sign test for global replication. The variance explained by each lead AAM signal in the DBDS was calculated using the formula $2 \times f (1 - f)\beta 2a$, where $f$ denotes the variant MAF and $\beta a$ is the effect estimate in additive models. The overall variance explained was calculated as the sum of individual variants.

### UK Biobank phenotype preparation

For downstream analyses in the UK Biobank, we derived an AAM variable using data from field 2714. To maximize sample size, individuals with missing or implausibly early or late 'first instance' AAM (<8 years old or >19 years old) were imputed using data from the next available instance (if plausible). We also derived two binary traits to represent abnormally early (precocious) and delayed puberty. Early puberty

was defined as AAM < 10 years old ($n = 1{,}321$). Delayed puberty was defined as AAM > 15 years old ($n = 10{,}530$). For comparison, participants reporting AAM at 12 or 13 years were controls ($n = 81{,}950$). All data analysis and visualization were conducted in R (version 4.2.1), unless otherwise stated.

### Rare variant associations with AAM

To identify gene-level rare variant associations with AAM, we performed an ExWAS analysis using whole-exome sequencing (WES) data on 222,283 UK Biobank women of European genetic-ancestry[77]. WES data processing and quality control were performed as described in ref. 30. Individual gene burden tests were performed by collapsing variants with MAF < 0.1% per gene according to their predicted functional consequences. We defined the following two functional categories of rare variants: (1) HC PTV annotated using VEP[78] and LOFTEE[79] and (2) DMG including HC PTVs plus missense variants with CADD score ≥25 (ref. 19). We analyzed a maximum of 17,885 protein-coding genes, each with at least ten rare allele carriers in either of the two variant categories ($P < 1.54 \times 10^{-6}$, 0.05/32,434 tests). Gene burden association tests were performed using BOLT-LMM[80]. The validity of the ExWAS analysis was indicated by the absence of significant association with the synonymous variant mask (Supplementary Fig. 1) and low exome-wide inflation scores ($\lambda$-PTV = 1.047 and $\lambda$-DMG = 1.047). Where applicable, protein domains were annotated using information from UniProt[81], and domain-level burden tests were then performed using linear models.

### Rare variant associations with other traits

We assessed the associations of any ExWAS AAM-associated genes in the UK Biobank with a range of related phenotypes – ANM (based on field 3581), BMI (based on field 21001), comparative body size age 10 (based on field 1687), adult height (based on field 50), comparative height age 10 (based on field 1697) and circulating IGF-1 concentrations (based on field 30770). We considered only the top AAM-associated variant mask for each gene. We also performed a similar lookup of these genes across a broader range of phenotypes using the AstraZeneca Portal[82].

### Rare variants in IHH panel app genes

We selected high-evidence green genes with an established monoallelic/X-linked mode of inheritance from the routine clinical investigation Genomics England gene panel for IHH. At the time of the study, this included the following four genes: *ANOS1*, *CHD7*, *FGF8* and *WDR11*. We performed a lookup of these genes in the UK Biobank WES data for AAM ($n = 222{,}283$) and VB ($n = 178{,}625$), considering only HC PTVs with MAF < 0.1%. We also extracted the phenotype of individual carriers. As in the ExWAS analysis, normal pubertal timing was defined in women as AAM between 10 and 15 years of age[1] and in men as AVB at an 'about average age' (UK Biobank data field 2385).

### PGS calculation

We calculated a genome-wide PGS for AAM using lassosum[31]. To keep PGS generation independent of PGS testing, we generated the PGS using our European-ancestry GWAS data excluding UK Biobank. We randomly selected 25,000 unrelated Europeans in the UK Biobank to generate the LD reference. The resulting PGS was standardized by subtracting the mean and dividing by the s.d.

We divided the PGS into 100 centiles and calculated the mean AAM for each PGS centile, as well as PGS centile-specific ORs for precocious or delayed AAM (as defined above) compared to individuals in the 50th centile of the PGS. We also calculated the mean PGS for each completed whole year of AAM.

We next tested whether the carriage of ExWAS AAM-associated rare variants modifies the influence of the PGS on AAM. We performed linear models that included interaction terms (PGS × rare variant carrier status) in the subsample of unrelated white-European UK Biobank

female participants with WES, PGS and AAM data ($n = 187,941$). To test for chance effects due to the low sample size, we randomly subsampled noncarriers to a sample size equivalent to that of carriers and compared this distribution of AAM to that observed in carriers.

A PGS comprising the 882 available lead AAM SNPs or their proxies (of the 935 independent AAM signals from the European analysis) was computed in 3,140 girls with available imputed GWAS data from the ALSPAC study. Linear or logistic regression models for continuous AAM, early AAM (−2 s.d., corresponding to <10.38 years) and delayed AAM (+2 s.d., corresponding to >14.95 years) were tested, controlling for the first 20 genetic principal components (PCs). Other models assessed the predictive performance of BMI at age 8 years and mother's AAM. Finally, a model including all predictors as covariables was calculated. The predictive performance of each model was evaluated by the $R^2$ metric for continuous AAM and by the AUROC for binary AAM outcomes.

### G2G pipeline

**Mapping GWAS signals to genes.** We performed signal selection on the two sets of AAM GWAS summary statistics, from the European-only and ancestry-combined meta-analyses. For each meta-analysis separately, we first filtered out all variants with MAF < 0.1%. The remaining variants were merged with allele information from the UK Biobank to provide the genomic sequence for any missing alleles. Genome-wide significant signals ($P < 5 \times 10^{-8}$) were selected initially based on proximity (in 1 Mb windows). Secondary signals at the same significance level ($P < 5 \times 10^{-8}$) within these windows were then identified using approximate conditional analysis (GCTA[83]). We generated an LD reference panel derived from 25,000 randomly selected white-European unrelated UK Biobank participants for GCTA and other downstream analyses, including analyses on the ancestry-combined data, due to the lack of approaches available for handling multi-ancestry LD reference panels. Secondary signals were defined if uncorrelated (in low LD, $r^2 < 0.05$) with the proximity-defined signals and if they did not show an overt change in their AAM association between baseline and conditional models (change in $\beta$ < 20% or change in $P$ value by less than four orders of magnitude). Primary and secondary AAM signals were further checked for pairwise LD within 10 Mb windows using PLINK (v1.90b6.18)[84], and only independent signals (in low LD, $r^2 < 0.05$) were retained, prioritizing distance-based signals in the case of linkage. Signal selection was performed first using the European-ancestry GWAS meta-analysis and then supplemented by any signals identified using the same process in the ancestry-combined meta-analysis that were uncorrelated ($r^2 < 0.05$) with any European-ancestry signal.

Independent GWAS AAM signals were examined for proximal genes, defined as those within 500 kb upstream or downstream of the gene's start or end sites, using the National Center for Biotechnology Information (NCBI) RefSeq gene map for GRCh37 (http://hgdownload.soe.ucsc.edu/goldenPath/hg19/database/).

**Colocalization with expression or pQTL data.** Tissue enrichment for GWAS associations was performed using LD score regression applied to tissue-specific expression (LDSC-SEG)[85] and tissue-specific annotations from the Genotype-Tissue Expression (GTEx; https://github.com/bulik/ldsc/wiki/Cell-type-specific-analyses). Significantly enriched tissues ($P < 0.05$) were then included in colocalization analyses with the tissue-specific and cross-tissue meta-analyzed GTEx eQTL data (v7 (ref. 65; available via https://gtexportal.org) and using the fixed-effects summary statistics for the latter), in addition to data from the eQTLGen[86] and Brain-eMeta[64] studies.

Including genomic variants with at least suggestive association with AAM (GWAS, $P < 5 \times 10^{-5}$), we applied summary data-based Mendelian randomization (MR) and heterogeneity in independent instruments (SMR-HEIDI, v0.68 (ref. 87)) and the approximate Bayes factor method in the R package 'coloc' (v5.1.0 (ref. 88)). For the former,

we considered gene expression to be influenced by the same GWAS AAM variant if the false discovery rate (FDR)-corrected SMR test $P < 0.05$ and HEIDI test $P > 0.001$. For the latter, genomic regions were defined as ±500 kb around each gene, and loci exhibiting an H4 PP > 0.75 were considered to show evidence of colocalization.

We also tested for colocalization between GWAS AAM variants and pQTLs from two datasets. First, we used pQTL data from the Fenland study[89], which includes data on 4,775 protein targets captured via the SomaScan v4 assay, measured in plasma from 10,708 European individuals. In addition, pQTL summary statistics from the UKB Pharma Proteomics Project[90] were used, which includes 2,923 protein targets captured via the Olink Explore 3072 proximity extension assay in up to 34,090 individuals of European ancestry. Colocalization was tested using the same procedure as mentioned above. It is important to note that colocalization analysis cannot determine causal relationships or the direction of causality between the two phenotypes.

**Mapping GWAS signals to enhancers and coding variants.** For genes proximal to (within 500 kb) GWAS AAM signals, we calculated genomic windows of high LD ($r^2 > 0.80$) around each signal and mapped these to the locations of known enhancers for the genes, using activity-by-contact (ABC) enhancer maps[34]. This was done across the 131 available cell/tissue types, and genes were matched to enhancers only in the tissues/cells where they were actively expressed.

We also checked whether GWAS AAM signals were in LD ($r^2 > 0.80$) with any coding variants within the paired genes and what the predicted consequence of those coding variants was, using SIFT[91] and POLYPHEN[92].

**Gene-level GWAS associations with AAM.** We performed a gene-level MAGMA analysis[35], which collapses common GWAS variants within each gene and calculates aggregate gene-level associations with the outcome trait, as described in ref. 35. To enhance the validity of this approach, we restricted the analysis to include only coding variants. Genes with FDR-corrected MAGMA $P < 0.05$ were considered associated with AAM.

Finally, we used the PoPS[36], which is a similarity-based gene prioritization method that uses cell-type-specific gene expression, biological pathways and protein–protein interactions to prioritize likely causal genes from GWAS data. At each locus, the gene with the numerically highest PoPS score was determined to be the PoPS-prioritized gene.

**Calculation of G2G scores.** From the abovementioned analyses, gene-level results were scored for each of the six sources as follows:

1. Closest gene: Gene proximity to a GWAS signal is a good predictor of causality[37]. The genes closest to each AAM signal (if also within 500 kb) were assigned. All genes with an intragenic signal were assigned as closest. The closest genes scored 1.5 points.
2. eQTL colocalization: Genes with evidence of eQTL colocalization via both SMR-HEIDI and coloc scored 1.5 points. Genes with evidence of colocalization via only one of these received 1.0 points. A further (1.0) point was assigned to genes if the most likely shared causal variant between eQTL and GWAS AAM was independent of the proximal GWAS signal ($r^2 < 0.05$).
3. pQTL colocalization: The same scoring as in 'eQTL colocalization' was applied to pQTL analyses.
4. Coding variants: As the evidence was overlapping for coding variant gene-level MAGMA analysis and signals correlated with coding variants, these analyses were scored concomitantly. Genes with an FDR-corrected MAGMA $P < 0.05$ were scored 0.5 points. Genes containing coding variants of deleterious or damaging predicted consequences in LD with GWAS AAM signals were scored 1.0 points, or only 0.5 points if the coding variants were predicted to be benign or tolerated.

5.  ABC enhancers: Genes targeted by enhancers that overlapped with or were correlated with GWAS AAM signals were scored 1.0 points.
6.  PoPS: PoPS-prioritized genes at each locus were scored 1.5 points.

G2G scores for each gene-signal pair were calculated as the sum of scores from these six sources. Genes that scored >0 points and were located within 500 kb of a GWAS AAM signal were considered further. To account for confounding due to large LD blocks, G2G scores were adjusted for signal LD window size (defined as the genomic distance containing variants with pairwise $r^2 > 0.50$ with the lead SNP) using linear regression models.

For genes proximal to more than one GWAS AAM signal (and hence with multiple G2G scores), the signal with the most concordant sources for that gene (highest residual G2G score) was retained and a further (1.0) point was added to reflect evidence from multiple signals. This resulted in a unique summarized G2G score for each included gene. To account for confounding due to gene size, G2G scores were further adjusted for gene length using linear regression models. The resulting residuals were considered to be the final G2G scores.

To prioritize likely causal AAM genes, all G2G-scored genes (that is, highlighted as potentially causal by at least one source) were ranked and also allocated a G2G centile position. In addition, the number of concordant predictors (sources) for each gene was noted (range: 1–6 sources). Finally, to reflect uncertainty due to multiple high-scoring genes for the same signal, genes were flagged if they were proximal (within 1 Mb) to other genes with a similar G2G score (within 0.5 points or greater and highlighted by at least the same number of sources).

**High-confidence AAM genes.** Independent GWAS signals from the all- and the European-ancestry meta-analyses were annotated with their top G2G scoring gene using corresponding GWAS data (that is, European analysis signals were annotated with genes from the European G2G, etc.). Genes implicated by at least two concordant sources were considered to be high-confidence AAM genes.

High-confidence AAM genes were functionally annotated using STRING[93]. Links to rare monogenic disorders were annotated from the Online Mendelian Inheritance in Man (OMIM) database (via OMIM; McKusick-Nathans Institute of Genetic Medicine, Johns Hopkins University; accessed November 2023, https://omim.org/). Finally, we used GTEx, a publicly available resource for tissue-specific gene expression, to lookup the tissue expression of 1,080 AAM genes highlighted by G2G[65].

**ZNF483 genome-wide binding analysis**
We used fGWAS (v.0.3.6 (ref. 32)), a hierarchical model for joint analysis of GWAS and genomic annotations, to test for enrichment of GWAS AAM signals among ZNF483 transcription factor binding sites. fGWAS models a maximum likelihood parameter estimate for the enrichment of a transcription factor (in this case, ZNF483). To perform this, we annotated the European-ancestry GWAS AAM summary statistics with the ZNF483 binding sites from the ENCODE ChIP–seq data derived from the human HepG2 cell line (ENCSR436PIH).

We also used SLDP (https://github.com/yakirr/sldp; ref. 33) to explore the directional effect of the ZNF483 function on AAM. We tested whether alleles that are predicted to increase the binding of ZNF483 have a combined tendency to increase or decrease AAM. SLDP requires signed LD profiles for ZNF483 binding, a signed background model and a reference panel in a SLDP-compatible format. We used a 1000 Genomes Phase 3 European reference panel containing approximately 10 million SNPs and 500 individuals.

**Clustering of AAM signals by early childhood body weight**
We analyzed repeated measurements of early childhood body weight from the MoBa cohort study[52,94] to investigate the relationship between early growth and puberty timing. Childhood body weight values were extracted from the study questionnaires for 12 different time points from birth to age 8 years using previously reported exclusion criteria[52]. Weight values were standardized and adjusted for sex and gestational age using the generalized additive model for location, scale and shape (GAMLSS; v5.1-7, www.gamlss.com) in R (v3.6.1) as previously reported[52] with the exception that a Box–Cox $t$ distribution was used to standardize body weight values (instead of the log-normal distribution used for BMI)[52]. GWAS for these traits was performed using BOLT-LMM (v2.3.4) as previously reported[52].

We performed MR analyses to assess the likely causal effects of AAM on childhood weight at each time point[95]. As instrumental variables (IVs), we used all 1,080 AAM-associated lead SNPs individually. As outcome data, we used childhood weight at the 12 time points. For SNPs with missing outcome data, we identified proxies within 1 Mb and $r^2 > 0.6$, choosing the variant with the highest $r^2$ value, using a random selection of 25,000 unrelated European-ancestry UK Biobank individuals for the LD reference. Genotypes at all variants were aligned to the AAM-increasing allele. We used IVW MR models, as this has the greatest statistical power[96].

Next, we stratified the 1,080 AAM lead SNPs by their effects on early childhood weight using a $k$-means clustering approach for longitudinal data[97]. We performed five different models with $k$-means for $k \in \{2,3,4,5,6\}$ clusters 20 times each. To find the optimal partition, we used the 'nearlyAll' option that uses several different initialization methods in alternation. As the assumption of homoscedasticity was not met, we used the Carolinski–Harabatz criterion, a nonparametric quality criterion, to derive the optimal number of clusters.

We then performed additional MR analyses, combining AAM signals within each identified cluster as IVs and, as the outcomes, childhood weight at each time point and also adult BMI (on 450,706 UK Biobank participants). We grouped together 'high early weight' and 'moderate early weight' AAM SNPs into a single IV to maximize power.

**Biological pathway enrichment analysis**
We performed gene-centric biological pathway enrichment analysis using g:Profiler (via the R client 'gprofiler2', v0.2.1 (ref. 53)). We used a filtered set of Gene Ontology (GO) pathways (accessed on 21 February 2023), focusing on GO:BP, KEGG and REACTOME, and restricted the analysis to those pathways with 1,000 genes or fewer, reasoning that these are more biologically specific. Pathway enrichment analyses were performed using the set of 665 high-confidence AAM genes and repeated when stratified by their effects on early childhood weight (see 'Clustering of AAM signals by early childhood body weight'). Pathways with Bonferroni-corrected $P < 0.05$ were considered to be associated with AAM.

As the pathways derived from overlapping sources, we clustered the AAM-associated pathways to aid interpretation. Clustering was based on shared AAM genes across pathways. We used a 'complete' clustering algorithm and a custom distance calculated as (one minus the proportion of the overlap between any two pathways relative to the pathway with the smaller overlap). Thus, between two pathways, a value of 0 indicates that all the shared AAM genes in the pathway with fewer genes are also enriched in the other pathway. To define clusters, we chose an arbitrary overlap value of 0.5, which indicates that pathways in the same cluster share 50% or more of their AAM genes.

Each pathway cluster was annotated by (1) the pathway with the most significant enrichment, (2) the pathway with the highest proportion of AAM genes, (3) the biological coherence of the pathways and (4) shared genes common to all included pathways. We considered that pathways were overlapping between the total AAM gene set and the two early weight subgroups if there were common pathways across either (1) the most significant pathway or (2) the pathway with the highest proportion of AAM-associated genes.

### Expression of AAM genes in GnRH neurons

We tested for enrichment of AAM-associated genes in RNA-seq data from embryonic GnRH mouse neurons[57]. All expressed genes were sorted into different expressional trajectories based on shared dynamic expression profiles across three developmental stages (early, intermediate or late; Zouaghi, et al.[57]). We tested for enrichment of AAM-associated genes (from our European-ancestry GWAS meta-analysis) in any identified trajectory, using MAGMA[35] with custom pathways. As a sensitivity test, we used Fisher's exact test to confirm the over-representation of AAM-associated genes within each trajectory.

### Colocalization of AAM signals with BMI and menopause

To explore the shared genetic architecture between AAM, ANM and adult BMI, we performed a colocalization analysis for each of the 1,080 AAM signals. ANM GWAS summary statistics were from reported ReproGen data on ~250,000 women of European ancestry[67]. Adult BMI GWAS summary statistics were derived from 450,706 individuals in the UK Biobank. For AAM signals with missing outcome GWAS data, we identified proxies within 1 Mb and with an $r^2 > 0.6$ using our 25,000-participant UK Biobank LD reference. We applied both Bonferroni correction ($P \le 0.05/1,080 = 4.6 \times 10^{-5}$) for association with the outcome trait, and a less stringent PP of colocalization PP > 0.5, due to the different priors for these hypothesis-driven analyses.

The same approach was applied in the opposite direction by testing ANM signals identified in the most recent ReproGen GWAS[67] for association with AAM. ANM signals were highlighted if they passed Bonferroni correction ($P \le 0.05/290 = 1.7 \times 10^{-4}$) for association with AAM. As ANM signals are well-established to be enriched for DNA DDR, we built a comprehensive list of DDR genes, integrating the following five different sources: (1) an expert-curated DDR gene list (broad DDR) from the laboratory of S. Jackson (this list encompasses a range of related pathways—DNA repair genes, broader DNA damage response genes (such as damage-induced chromatin remodeling, transcription regulation or cell cycle checkpoint induction)) and general maintenance of genome stability (such as genes involved in DNA replication); (2) a second expert-curated list previously reported[67] (assembled by J. Perry, E. Hoffmann and A. Murray); (3) genes listed in the REACTOME[98] 'DNA repair' pathway (R-HSA-73894); (4) genes listed in the GO 'DNA repair' pathway (GO:0006281) and (5) genes listed in the GO[99] 'cellular response to DNA damage stimulus' (GO:0006974).

### GPR83–MC3R interaction

**Brain-expressed GPCRs.** We tested whether any brain-expressed GPCRs were implicated by GWAS AAM associations (G2G scores). We tested the following list of brain-expressed GPCRs[59]: *ACKR1, ACKR2, ACKR3, ACKR4, ADRB1, ADRB2, ADRB3, AGTR1, AGTR2, BRS3, C5AR1, C5AR2, CALCR, CASR, CCKAR, CCR1, CCR10, CCR2, CCR3, CCR4, CCR5, CCR6, CCR7, CCR9, CCRL2, CNR1, CNR2, CXCR1, CXCR2, CXCR3, CXCR4, CXCR6, DRD1, DRD2, DRD3, DRD4, DRD5, EDNRA, EDNRB, FFAR1, FFAR2, FFAR3, FFAR4, FPR1, FPR2, FPR3, FSHR, GALR1, GALR2, GALR3, GHRHR, GHSR, GIPR, GLP1R, GLP2R, GNRHR, GPER1, GPR1, GPR12, GPR15, GPR17, GPR18, GPR19, GPR20, GPR22, GPR25, GPR26, GPR27, GPR3, GPR34, GPR35, GPR37, GPR39, GPR4, GPR42, GPR45, GPR52, GPR55, GPR6, GPR61, GPR62, GPR63, GPR75, GPR78, GPR82, GPR83, GPR84, GPR85, GPR87, GPR88, GRM1, GRM2, GRM3, GRM4, GRM5, GRM6, GRM7, GRM8, GRPR, HCAR1, HCAR2, HCAR3, HRH1, HRH2, HRH3, HRH4, LGR4, LGR5, LGR6, LPAR1, LPAR2, LPAR3, LPAR4, LPAR5, LPAR6, MC3R, MC4R, MC5R, MCHR1, MCHR2, NMBR, NMUR1, NMUR2, NPSR1, NPY1R, NPY2R, NPY4R, NPY5R, NPY6R, OXER1, OXGR1, P2RY1, P2RY2, P2RY4, P2RY6, P2RY8, PRLHR, PTAFR, PTH1R, PTH2R, QRFPR, RGR, RXFP1, RXFP2, S1PR1, S1PR2, S1PR3, S1PR4, S1PR5, SCTR, SSR1, SSR2, SSR3, SSR4, TSHR, VIPR1, VIPR2, VN1R1, VN1R2, VN1R5* and *XCR1*. For any GPCR scored by our G2G AAM pipeline, colocalization was tested between GWAS signals for AAM and adult BMI (colocalization methods as described above).

**Cell culture and transfection.** To investigate the effect of *GPR83* on MC3R signaling, we performed in vitro assays in transiently transfected HEK293 cells maintained in Dulbecco's modified eagle medium (high glucose DMEM; Gibco, 41965) supplemented with 10% fetal bovine serum (Gibco, 10270), 1% GlutaMAX (100×; Gibco, 35050) and 100 units per ml penicillin and 100 mg ml$^{-1}$ streptomycin (Sigma-Aldrich, P0781). Cells were incubated at 37 °C in humidified air containing 5% $CO_2$, and transfections were performed using Lipofectamine 2000 (Gibco, 11668) in serum-free Opti-MEM I medium (Gibco, 31985), according to the manufacturer's protocols. The plasmids used encode the C-FLAG-tagged human *GPR83* WT (NM_016540.4) or N-FLAG-tagged human MC3R WT (NM_019888.3) ligated into pcDNA3.1(+) (Invitrogen).

**BRET to measure dimerization.** Heterodimerization between *GPR83* and *MC3R* was quantified using BRET1 in titration configuration. Briefly, 12,000 HEK293 cells seeded in 96-well plates were transfected with a constant dose of MC3R-RlucII plasmid (0.5 ng per well) and increasing doses of *GPR83*-Venus plasmids, or soluble (s) Venus as negative control. All conditions were topped up with an empty vector (pcDNA3.1 (+)) to a total of 100 ng of plasmid per well. Twenty-four hours post-transfection, cells were washed once with Tyrode's buffer and total Venus fluorescence was measured in a Spark 10M microplate reader (Tecan) using monochromators (excitation 485 ± 20 nm and emission 535 ± 20 nm). BRET was quantified 10 min after the addition of coelenterazine H (NanoLight Technology; 2.5 mM). netBRET was calculated as (absorbance at 533 ± 25 nm/absorbance at 480 ± 40 nm) − (background (absorbance at 533 ± 25 nm/absorbance at 480 ± 40 nm)), with the background corresponding to the signal in cells expressing the RlucII protomer alone under similar conditions. Data on the $x$ axis represent the ratio between acceptor (Venus) fluorescence and donor (RlucII) luminescence. Representative data are from four independent experiments.

**Time-resolved cAMP assay.** Measurement of ligand-induced cAMP generation in HEK293 cells transiently expressing either *MC3R* or both *MC3R* and *GPR83* was performed using the GloSensor cAMP biosensor (Promega), according to the manufacturer's protocol. Briefly, 12,000 cells were seeded in white 96-well poly-D-lysine-coated plates. After 24 h, cells were transfected with both 100 ng per well of pGloSensorTM-20F cAMP plasmid (Promega, E1171) and 30 ng per well of each plasmid encoding either MC3R or MC3R and GPR83, using Lipofectamine 2000 (Gibco, 11668). All conditions were topped up with an empty vector (pcDNA3.1 (+)) to a total of 160 ng of plasmid per well. The day after transfection, cell media were replaced by 90 ml of fresh DMEM with 2% vol/vol GloSensorTM cAMP reagent (Promega, E1290) and incubated for 120 min at 37 °C. Firefly luciferase activity was measured at 37 °C and 5% $CO_2$ using a Spark 10M microplate reader (Tecan). After initial measurement of the baseline signal for 10 min (30 s intervals), cells were stimulated with 10 ml of 10× stock solution of the MC3R agonist NDP-aMSH (final concentration was 1 mM), and real-time chemiluminescent signals were quantified for 25 min (30 s intervals). In each experiment, a negative control using mock-transfected cells (empty pcDNA3.1(+) plasmid) was assayed. The area under the curve (AUC) was calculated for each cAMP production curve considering the total peak area above the baseline calculated as the average signal for mock pcDNA3.1(+)-transfected cells. For data normalization, the AUC from mock-transfected cells was set as 0 and the AUC from WT MC3R was set as 100%. The results are from six independent experiments.

**Genetic epistasis between GPR83 and MC3R.** To corroborate the in vitro interaction, we tested for evidence of a specific epistatic interaction between AAM GWAS signals at *GPR83* (rs592068-C) and *MC3R* (rs3746619-A). We extracted genotypes for these SNPs in white-European unrelated UK Biobank women (204,303). After adjusting AAM for standard covariates (GWAS chip, age, sex and PC1-10),

we modeled the interaction between genotype dosages at the two signals using a linear model.

## Statistics and reproducibility

For GWAS discovery analyses, we meta-analyzed data from all large-scale biobanks and consortia with puberty data ($n = 799,845$). For WES discovery analyses, we used the full available sample with available data in the UK Biobank ($n = 222,283$). Only individuals failing standard genotyping quality control parameters defined in the individual studies or missing genotype, phenotype, or covariate data were excluded from the analysis. This decision was made before performing any downstream analysis. Where described, sensitivity analyses were performed in subsets of the UK Biobank cohort, with exclusions of related individuals and/or non-European-ancestry individuals. We replicated our GWAS findings using menarche association data from the Danish Blood Donors study ($n = 35,467$) and the ALSPAC study ($n = 3,140$). Menarche ExWAS and GWAS data were also replicated using data on voice-breaking ($n = 178,625$ and up to $n = 205,354$ accordingly). All attempted replications have been reported in the manuscript without exception.

The principle exposure in this study is naturally occurring genetic variants, meaning that we were unable to randomize the individuals in the study. To account for possible confounding, we used a linear mixed model and adjusted for technical and demographic covariates. Blinding is not applicable to this study, as it is a GWAS of common and rare genetic variation and not a randomized controlled trial. We did not deliver any interventions to the participants in this study.

### Reporting summary

Further information on research design is available in the Nature Portfolio Reporting Summary linked to this article.

## Data availability

Cohorts should be contacted individually for access to their raw data, which are not publicly available as they contain information that could compromise the privacy of their research participants. UK Biobank data are available on the application (https://www.ukbiobank.ac.uk/enable-your-research/register). Summary statistics from the European-only and ancestry-combined meta-analyses, excluding 23andMe, are available via https://doi.org/10.17863/CAM.107943. Access to the full summary statistics, including 23andMe results, can be obtained from 23andMe after completion of a Data Transfer Agreement (https://research.23andme.com/dataset-access/). We used the NCBI RefSeq gene map for GRCh37, which is available via http://hgdownload.soe.ucsc.edu/goldenPath/hg19/database/. GTEx eQTL V7 data were used and are available via https://gtexportal.org. Genes linked to rare monogenic disorders were annotated from the OMIM database (via OMIM; McKusick-Nathans Institute of Genetic Medicine, Johns Hopkins University; accessed November 2023, https://omim.org/).

## Code availability

The meta-analysis was performed using a fixed-effects IVW model in METAL (https://github.com/statgen/METAL, 25 March 2011). Conditionally independent signals were identified using GCTA (v1.92.0). All LD calculations were performed in PLINK (https://www.cog-genomics.org/plink2/, v1.90b6.18). LDSC-SEG (https://github.com/bulik/ldsc, v1.0.1) was used to perform tissue enrichment analyses. eQTL and pQTL colocalization analyses were performed using the R package 'coloc' (v5.1.087) and SMR-HEIDI (v0.6886). We used MAGMA (v1.09), PoPS (https://github.com/FinucaneLab/pops, v0.2), lassosum (https://github.com/tshmak/lassosum, v0.4.5), fGWAS (https://github.com/joepickrell/fgwas, v0.3.632), SLDP (https://github.com/yakirr/sldp), generalized additive model for location, scale and shape (GAMLSS; v5.1-7, via www.gamlss.com) in R (v3.6.1) and STRING (https://string-db.org/, v12.0). Pathway enrichment analysis were performed using g:Profiler (via the R client 'gprofiler2', v0.2.1). Plots were created using R (v4.2.1)

using ggplot2 (v3.3.6) and the 'Zissou1' palette from the wesanderson R package (v0.3.6). Code for WES data processing and association testing is available on GitHub (https://github.com/mrcepid-rap/mrcepid-runassociationtesting).

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

## Acknowledgements

This research was supported by the UK Medical Research Council (MRC; Unit program MC_UU_00006/2) and has been conducted using the UK Biobank Resource under application 9905. Other study-specific acknowledgements can be found in the Supplementary Information.

## Author contributions

K.A.K., L.R.K., F.R.D., J.R.B.P. and K.K.O. wrote the manuscript. K.A.K., L.R.K., S. Stankovic, M.V., E. Bratland, A.S.B., E.J.G., M.Y., N.Y., D.M., S.J., F.R.D., J.R.B.P. and K.K.O. contributed to data analysis and interpretation. E.M.d.O., K.M.P., I.H.D. and I.S.F. contributed the in vitro GPR83–MC3R data. A. Messina, Y. Zouaghi, F.S. and N.P. contributed to the mouse embryonic GnRH neuron RNA-seq data. All authors contributed data to the study, revised, critically reviewed and approved the final version of the manuscript.

## Competing interests

J.R.B.P. and E.J.G. are employees/shareholders of Insmed. J.R.B.P. also receives research funding from GSK and consultancy fees from WW International. Y. Zhao is a UK University worker at GSK. D.L.C. is an employee and shareholder of GSK. J.P.B. is employed by GSK. I.S.F. has consulted for a number of companies developing weight loss drugs including Eli Lilly, Novo Nordisk and Rhythm Pharmaceuticals. D.J.T. is employed by Genomics PLC. E.T. is employed by Pfizer. D.A.L. has received support from Roche Diagnostics and Medtronic for work unrelated to the research in this paper. T.D.S. is cofounder and stakeholder of Zoe Global. P.A.F. conducts research funded by Amgen, Novartis and Pfizer, and he received Honoraria from Roche, Novartis and Pfizer. M.W.B. conducts research funded by Amgen, Novartis and Pfizer. M.I.M. is currently an employee of Genentech and a holder of Roche stock. The other authors declare no competing interests.

## Additional information

**Correspondence and requests for materials** should be addressed to John R. B. Perry.

# Reporting Summary

## Statistics

For all statistical analyses, confirm that the following items are present in the figure legend, table legend, main text, or Methods section.

| n/a | Confirmed | |
|---|---|---|
| ☐ | ☒ | The exact sample size (*n*) for each experimental group/condition, given as a discrete number and unit of measurement |
| ☐ | ☒ | A statement on whether measurements were taken from distinct samples or whether the same sample was measured repeatedly |
| ☐ | ☒ | The statistical test(s) used AND whether they are one- or two-sided *Only common tests should be described solely by name; describe more complex techniques in the Methods section.* |
| ☐ | ☒ | A description of all covariates tested |
| ☐ | ☒ | A description of any assumptions or corrections, such as tests of normality and adjustment for multiple comparisons |
| ☐ | ☒ | A full description of the statistical parameters including central tendency (e.g. means) or other basic estimates (e.g. regression coefficient) AND variation (e.g. standard deviation) or associated estimates of uncertainty (e.g. confidence intervals) |
| ☐ | ☒ | For null hypothesis testing, the test statistic (e.g. *F*, *t*, *r*) with confidence intervals, effect sizes, degrees of freedom and *P* value noted *Give P values as exact values whenever suitable.* |
| ☐ | ☒ | For Bayesian analysis, information on the choice of priors and Markov chain Monte Carlo settings |
| ☐ | ☒ | For hierarchical and complex designs, identification of the appropriate level for tests and full reporting of outcomes |
| ☐ | ☒ | Estimates of effect sizes (e.g. Cohen's *d*, Pearson's *r*), indicating how they were calculated |

*Our web collection on statistics for biologists contains articles on many of the points above.*

## Software and code

Policy information about availability of computer code

| Data collection | No software used |
|---|---|
| Data analysis | The meta-analysis was performed using a fixed-effects inverse-variance-weighted model in METAL (https://github.com/statgen/METAL, 2011/03/25). Conditionally independent signals were identified using GCTA (version 1.92.0). All LD calculations were performed in plink (https://www.cog-genomics.org/plink2/, version 1.90b6.18). LDSC-SEG (https://github.com/bulik/ldsc, version 1.0.1) was used to perform tissue enrichment analyses. e- and pQTL colocalisation analyses were performed using the R package "coloc" (version 5.1.087) and SMR-HEIDI (version 0.6886). We used MAGMA (version 1.09), PoPS (https://github.com/FinucaneLab/pops, version 0.2), lassosum (https://github.com/tshmak/lassosum, version 0.4.5), fGWAS (https://github.com/joepickrell/fgwas, version 0.3.632), Signed LD Profile regression (SLDP, https://github.com/yakirr/sldp), generalized additive model for location, scale and shape (GAMLSS, version 5.1-7, via www.gamlss.com) in R (v3.6.1), and STRING (https://string-db.org/, version 12.0). Pathway enrichment analysis were performed using g:Profiler (via the R client "gprofiler2", version 0.2.1). Plots were created using R (version 4.2.1) using ggplot2 (3.3.6) and the "Zissou1" palette from the wesanderson R package (version 0.3.6). Code for WES data processing and association testing is available on GitHub (https://github.com/mrcepid-rap/mrcepid-runassociationtesting). |

For manuscripts utilizing custom algorithms or software that are central to the research but not yet described in published literature, software must be made available to editors and reviewers. We strongly encourage code deposition in a community repository (e.g. GitHub). See the Nature Portfolio guidelines for submitting code & software for further information.

## Data

Policy information about <u>availability of data</u>

All manuscripts must include a <u>data availability statement</u>. This statement should provide the following information, where applicable:

- Accession codes, unique identifiers, or web links for publicly available datasets
- A description of any restrictions on data availability
- For clinical datasets or third party data, please ensure that the statement adheres to our <u>policy</u>

> Cohorts should be contacted individually for access to their raw data. UK Biobank data are available on application (https://www.ukbiobank.ac.uk/enable-your-research/register). Summary statistics from the meta-analyses, excluding 23andMe, are available via https://doi.org/10.17863/CAM.107943. Access to the full summary statistics including 23andMe results, can be obtained from 23andMe after completion of a Data Transfer Agreement (https://research.23andme.com/dataset-access/).
> We used the NCBI RefSeq gene map for GRCh37 which is available via http://hgdownload.soe.ucsc.edu/goldenPath/hg19/database/. GTEx eQTL data was used (V7) and is available via https://gtexportal.org. Genes linked to rare monogenic disorders were annotated from the Online Mendelian Inheritance in Man (OMIM) database (via Online Mendelian Inheritance in Man, OMIM®. McKusick-Nathans Institute of Genetic Medicine, Johns Hopkins University (Baltimore, MD), accessed November 2023. World Wide Web URL: https://omim.org/).

## Research involving human participants, their data, or biological material

Policy information about studies with <u>human participants or human data</u>. See also policy information about <u>sex, gender (identity/presentation), and sexual orientation</u> and <u>race, ethnicity and racism</u>.

| | |
|---|---|
| Reporting on sex and gender | All analyses are disaggregated by sex.<br>All contributing studies had research ethics approval and collected informed written consent. |
| Reporting on race, ethnicity, or other socially relevant groupings | Study samples were defined by genetic ancestry. |
| Population characteristics | Described in Supplementary Table 1 "Cohorts" |
| Recruitment | This study is a meta-analysis of data from a number of sources. Recruitment varied across the different studies that contributed data. |
| Ethics oversight | UK Biobank data has approval from the North West Multi-centre Research Ethics Committee (MREC) as a Research Tissue Bank (RTB). 23andMe research participants provided informed consent and volunteered to participate in the research online under a protocol approved by the external AAHRPP-accredited IRB, Ethical & Independent (E&I) Review Services. Each of the other individual studies that contributed data has their own ethical approval from the relevant boards. |

Note that full information on the approval of the study protocol must also be provided in the manuscript.

# Field-specific reporting

Please select the one below that is the best fit for your research. If you are not sure, read the appropriate sections before making your selection.

☒ Life sciences  ☐ Behavioural & social sciences  ☐ Ecological, evolutionary & environmental sciences

For a reference copy of the document with all sections, see <u>nature.com/documents/nr-reporting-summary-flat.pdf</u>

# Life sciences study design

All studies must disclose on these points even when the disclosure is negative.

| | |
|---|---|
| Sample size | For GWAS discovery analyses, we meta-analysed data from all large-scale Biobanks and consortia with puberty data (n=799,845). For whole exome sequencing discovery analyses, we used the full available sample with available data in UK Biobank (n=222,283). |
| Data exclusions | Only individuals failing standard genotyping quality control parameters defined in the individual studies or missing genotype, phenotype, or covariate data were excluded from analysis. This decision was made prior to performing any downstream analysis. Where described, sensitivity analyses were performed in subsets of the UK Biobank cohort, with exclusions of related individuals and/or non-European ancestry individuals. |
| Replication | We replicated our GWAS findings using menarche association data from the Danish Blood Donors study (n=35,467) and the ALSPAC study (n=3,140). Menarche ExWAS and GWAS data were also replicated using data on voice-breaking (n=178,625 and up to n=205,354, accordingly). All attempted replication has been reported in the manuscript without exception. |
| Randomization | The principle exposure in this study is naturally occurring genetic variants, meaning that we were unable to randomize the individuals in the study. To account for possible confounding, we used a linear mixed model and adjusted for technical and demographic covariates. |

| Blinding | Blinding is not applicable to this study, as it is a genome-wide association study of common and rare genetic variation and not a randomized controlled trial. We did not deliver any intervention to the participants in this study. |
|---|---|

# Reporting for specific materials, systems and methods

We require information from authors about some types of materials, experimental systems and methods used in many studies. Here, indicate whether each material, system or method listed is relevant to your study. If you are not sure if a list item applies to your research, read the appropriate section before selecting a response.

## Materials & experimental systems

| n/a | Involved in the study |
|---|---|
| ☒ | ☐ Antibodies |
| ☐ | ☒ Eukaryotic cell lines |
| ☒ | ☐ Palaeontology and archaeology |
| ☒ | ☐ Animals and other organisms |
| ☒ | ☐ Clinical data |
| ☒ | ☐ Dual use research of concern |
| ☒ | ☐ Plants |

## Methods

| n/a | Involved in the study |
|---|---|
| ☒ | ☐ ChIP-seq |
| ☒ | ☐ Flow cytometry |
| ☒ | ☐ MRI-based neuroimaging |

## Eukaryotic cell lines

Policy information about cell lines and Sex and Gender in Research

| Cell line source(s) | HEK293 cells were kindly provided by Professor Michel Bouvier, Universite de Montreal, Canada. |
|---|---|
| Authentication | HEK293 cells were authenticated via GENETICA Genotypes Analysis in May 2019, showing 97% match when compared to the reference profile ATCC sequence. |
| Mycoplasma contamination | HEK293 cells tested negative for mycoplasma contamination using MycoProbe Mycoplasma Detection Kit (CUL001B, R&D Systems). |
| Commonly misidentified lines (See ICLAC register) | This is not a commonly misidentified cell line, as listed on the ICLAC register (version 12). |

## Plants

| Seed stocks | NA |
|---|---|
| Novel plant genotypes | NA |
| Authentication | NA |

