## [Peer Review File · Nature Genetics]

Peer Review Information

Manuscript Title: Understanding the genetic complexity of puberty timing across the allele frequency spectrum

Corresponding author name(s): Professor John Perry

Editorial Notes:

Transferred manuscripts This document only contains reviewer comments, rebuttal and decision letters for versions considered at Nature Genetics.

Reviewer Comments & Decisions:

Decision Letter, initial version:

13th Nov 2023

Dear Professor Ong,

First, please accept my most sincere (and mortified) apologies at the delay in returning this decision to you. I am so sorry for keeping you waiting for so long, and can't tell you how grateful I am for your patience.

Your Article, "Understanding the genetic complexity of puberty timing across the allele frequency spectrum" has now been seen by 3 referees. You will see from their comments below that while they find your work of interest, some important points are raised. We are interested in the possibility of publishing your study in Nature Genetics, but would like to consider your response to these concerns in the form of a revised manuscript before we make a final decision on publication.

We therefore invite you to revise your manuscript taking into account all reviewer comments. Please highlight all changes in the manuscript text file. At this stage we will need you to upload a copy of the manuscript in MS Word .docx or similar editable format.

*2) If you have not done so already please begin to revise your manuscript so that it conforms to our Article format instructions, available here.
Refer also to any guidelines provided in this letter.

Please be aware of our guidelines on digital image standards.

[redacted]

We hope to receive your revised manuscript within four to eight weeks. If you cannot send it within this time, please let us know.

Sincerely,

Safia Danovi
Editor
Nature Genetics

Referee expertise:

Referee #1: GWAS, women's health

Referee #2: GWAS, statistical genetics

Referee #3: GWAS, reproductive endocrinology

Reviewers' Comments:

Reviewer #1:

Remarks to the Author:

Multi-ancestry GWAS meta-analysis of 800K females identified 1,080 common genome-wide significant variants associated with age at menarche (AAM), a marker of puberty timing in females. Variations in age of menarche is a well established risk factors for wide-range of conditions. 11% of trait variance is explained by these variants. In addition these are complemented by exome-wide association analysis from 220K females collapsing rare variants in each gene and performing gene-burden tests which identifies 6 associated genes. The authors also employed extensive in silico gene mapping approaches to identify the genes regulated by the identified variants which results in identification of 660 potentially relevant genes. They propose some very interesting hypotheses based on the implicated genes linking puberty timing with various disease risks dependent and independent of body-size.

This study is more than doubling the sample size from previous GWAS for AAM that was also previously limited to only European ancestry and now incorporating of their sample from East Asian ancestry. This has resulted in significant boost in power to detect many novel variants for the trait (Variance explained increases from ~7.4% to 11%; additional 145 signals are identified from all-ancestry analysis in addition to European only analyses that identified 935 signals). In addition, they present the first large-scale effort to assess the rare spectrum of variation for AAM utilising gene burden tests by collapsing rare variants which revealed 6 genes associated with AAM. Looking up these genes in voice break data in males is an interesting way suggesting some potentially sex-specific differences. Their GWAS to gene pipeline to map identified signals to genes is comprehensive and utilises multiple data sources to identified in silico functional evidence that colocalise with the signals. Then very interesting and with available data on body weight from age birth to age 8 years, they checked whether the identified 1080 AAM signals had an effect on childhood weight that revealed 3 different trajectories where >50% of AAM signals had no early weight gain trajectory and 44% with moderate weight gain and 1% with high early weight gain. This is very valuable in terms of dissecting those signals with primary effects on puberty timing or early weight gain. In addition to standard pathway enrichment analysis of the identified AAM genes, they have explored an unpublished RNAseq-based differential expression dataset comparing differences in expression between embryonic migration stages of embryonic GnRH neurone in mouse. This works seems very interesting but I think deserves a line on 708 to state relevance of the GnRH neurons to AAM. They show in vitro evidence for interaction between MC3R and GPR83 and also statistical genetic evidence for epistasis between the associated SNPS in both genes. This is very exciting as it enlightens their previous work

illustrating MC3R loss of function association with delayed puberty to be enhanced by GPR83 expression. Last but not least, results showing that there are 33 shared signals between AAM and age at menopause and their potential links with later on onset conditions life is very informing. This is a huge amount of work shading light on novel mechanisms underlying the roles of genetic determinant of start and end of reproductive lifespan. The methodology and quality of presentation is great. Conclusions are sensible. Publication of this manuscript in Nature Genetics will make sure these reach the wide scientific audience it highly deserves and will benefit.

Some specific comments:

(1) It would be informative to see the allele frequencies of the identified 1080 variants in each ancestry separately, e.i. in ST2 adding allele frequency for each variants for European ancestry and East Asian ancestry separately.

(2) Authors may considering estimating potential heterogeneity contributed via differences between the contributing studies (ascertainment (e.g. cancer datasets, biobanks), ancestry (European vs. East Asian)) in the identified genome-wide signals via meta-regression.

(3) In exome-wide association results, authors report 6 genes associated significantly with AAM. However, from line 562, they start talking about confirmation of genes from a set of seven. Needs to be clarified where the additional gene is coming from or it a typo.

Reviewer #2:

Remarks to the Author:

Ong and colleagues present findings from the largest GWAS of AAM to date, expanding over previous efforts by aggregating European and East Asian ancestry studies. The authors identify more than 1000 signals of association that explain 11% of AAM variance and are associated with delayed and precocious pubertal development in the extremes of the polygenic score distribution. Analyses of exome sequence data highlight genes with a burden of PTV/DMG variants associated with AAM, which accounted for the polygenic risk of common variants in carriers. A total of 660 high-confidence genes were identified using a variant to gene mapping strategy that brought together support from diverse sources, including proximity and colocalization with molecular QTLs. Downstream analyses highlighted the biological complexity of AAM, and highlight clusters of signals that reflect association with early weight gain and BMI later in life.

The manuscript brings together a broad range of diverse analyses and data resources to better understand the genetic contribution to AAM, and represents a considerable advance over previous AAM GWAS efforts. The investigation of the contribution of common and rare variants to polygenic risk was a highlight. The manuscript is generally clearly written and straightforward to follow. My review focuses on the genetic analyses, which are mostly robust, although I do have some concerns and requested clarifications, as outlined below.

Line 502. Presumably the first four of the five strata are European ancestry. This would be worth emphasizing.

Line 511. I could not follow the rationale for signal selection. I can understand using UK Biobank as a reference for LD, but GCTA would normally be used to then identify distinct signals at some pre-

defined significance threshold. I think that a genome-wide significance threshold (5×10^{-8}) was used, but it is not clear from the Methods. The criteria for signal selection then seem very complex, based on changes in the magnitude of the P-value after conditioning and R^2 , and I could not see any justification for this. Presumably this is LD r^2 , as opposed to variance explained, but based on UK Biobank European ancestry data? I was then not clear how signals identified in the European ancestry meta-analysis from a conditional analysis could be meta-analysed with the East Asian ancestry data (which will not necessarily have been conditioned on the same set of variants). Are the summary statistics from Table ST2 from unconditional or conditional meta-analysis? Are all signals genome-wide significant in conditional analysis?

Line 511. It wasn't clear how the strategy for signal selection would translate into the all-ancestry meta-analysis. For example, there could be a variant that is in strong LD with a signal identified in the European ancestry meta-analysis based on East Asian ancestry LD, but not European LD (captured in UK Biobank), and it is not clear to me that this would constitute an independent signal. Given that the authors state that their aim is to identify signals that are homogeneous across the two strata, I would suggest focussing on clumping genome-wide significant variants from the all-ancestry meta-analysis, requiring independent signals to have $r^2 < 0.05$ in both European and East Asian ancestry reference data. It would recommend NOT referring to the meta-analysis of European and East Asian ancestry data as "all ancestry" as the meta-analysis only brings together data from two ancestries.

Line 520. I would be more interested in understanding evidence for heterogeneity in effects between the European and East Asian ancestry findings. This could be done by comparing effect sizes from the European ancestry meta-analysis with those from the East Asian ancestry meta-analysis. Also of interest is whether the 935 European ancestry signals validate in the East Asian ancestry meta-analysis.

Line 530. The overlap of signals between AAM and AVB in UK Biobank and 23andMe looks convincing. I wonder why the results from these two resources were not meta-analysed to maximise power, and a look-up of effects in the meta-analysis conducted. Whilst the concordance in the direction of effect is interesting, I wonder if the more powerful test would be $P < 0.05$ and concordant direction of effect. Only 2.5% of signals would be expected to meet these two criteria by chance, and I think the support for an enrichment of signals between the two traits would be more convincing.

Line 562. I was confused what the "seven" genes referred to. Are these the six at exome-wide significance and then KDM5B? I'd be tempted to drop the report of KDM5B since this the near significant findings is actually from a different functional variant category, and was not associated with AVB.

Line 672. The clustering of AAM signals according to early weight gain is very interesting. Have the authors considered testing the effects of these different clusters of SNPs for association with some of the diseases that are associated with adult-onset diseases, by using partitioned polygenic scores, for example?

Line 1006. Many of the signal to gene mappings were based on LD. How was LD assessed, particularly if signals were identified in the multi-ancestry meta-analysis.

Line 1032. It would be useful to report the results of the tissue enrichment analysis in the main text, even in just a couple of sentences. It was difficult to dig through the supplementary material to

identify the tissues. Presumably, from Supplementary Figure 9, the enriched tissues were adrenal, brain and pituitary?

Line 1043. What was the justification for the use of PP H4 >0.75 for colocalization from coloc for eQTLs and pQTLs? This doesn't seem to be consistent with what is reported in the main text (PP H4 >0.5)?

Line 1068. The calculation of the G2G scores seem rather arbitrary. What was the justification for the different weightings applied across the different sources of evidence?

Reviewer #3:

Remarks to the Author:

The study "Understanding the genetic complexity of puberty timing across the allele frequency spectrum" by Kentistou & Kaisinger et al aims to elucidate the biology of pubertal timing by performing multi-ancestry genetic analyses in 800,000 women, complemented by exome sequencing analyses in 220,000 women, in silico variant-to-gene mapping, and integration with gene expression data from mouse embryonic GnRH neurons.

As a result, the authors identify over 1000 independent signals explaining 11% of menarche timing, characterise rare loss of function variants that affect the trait and propose novel biological mechanisms for pubertal timing in females.

Overall, the manuscript is well-written, clear and concise. The work represents another important step forward in understanding the genetic mechanism governing pubertal timing that has also important implications for overall health and wellbeing. It is especially positive to see the authors have included several non-European datasets (the China Kadoorie Biobank, the Biobank Japan, the Korean Genome and Epidemiology Study) to improve diversity in genetic studies.

The selected methods and analyses are appropriate and standard in the field and have mostly been described in sufficient detail to allow replication. The results of these analyses also support the drawn conclusions and nicely informative figures have been used to communicate and summarise the most results.

I was wondering about the conclusion on page 20, lines 832-835 "Together, these insights shed light on mechanisms, including early weight gain and adiposity, hormone secretion and response, and cellular susceptibility to DNA damage, that potentially explain the widely reported relationships between earlier puberty timing and higher risks of later life mortality, metabolic disease, and cancer.", drawn from the results that show some of the genetic signals shared between menarche and menopause map to DNA damage response genes or to the HPG axis. In the authors' opinion, regarding the reported associations between puberty timing and cancer risk and in the light on their new results - is the association mostly due to the extended hormonal exposure or due to shared DDR mechanisms (or both)? Could the genetic data somehow be used to test which of these biological mechanisms is driving the association?

Minor questions:

1) Is there any overlap between the breast cancer association consortium and ovarian cancer association consortium samples and rerogen/ukbb/23andme data? Could it possibly affect the results

somehow?

2) Were there any ancestry-specific GWAS findings worth highlighting?

Author Rebuttal to Initial comments

NG-A62753R

Response to reviewers' comments

Response: We are grateful to the reviewers for their time and helpful comments. In addition to the responses to each comment below, we note that we have updated our variant to gene mapping pipeline by integrating newly released data from the UK Biobank Pharma Proteomics Project (released in late 2023). This introduced a minor change to the number of “high-confidence” genes (665 instead of 660) and without significant change to the genes or processes highlighted by the downstream analyses.

Reviewers' comments are in blue below. All new changes to the manuscript and display items are in red.

Referee #1: GWAS, women's health

Multi-ancestry GWAS meta-analysis of 800K females identified 1,080 common genome-wide significant variants associated with age at menarche (AAM), a marker of puberty timing in females. Variations in age of menarche is a well established risk factors for wide-range of conditions. 11% of trait variance is explained by these variants. In addition these are complemented by exome-wide association analysis from 220K females collapsing rare variants in each gene and performing gene-burden tests which identifies 6 associated genes. The authors also employed extensive in silico gene mapping approaches to identify the genes regulated by the identified variants which results in identification of 660 potentially relevant genes. They propose some very interesting hypotheses based on the implicated genes linking puberty timing with various disease risks dependent and independent of body-size.

This study is more than doubling the sample size from previous GWAS for AAM that was also previously limited to only European ancestry and now incorporating of their sample from East Asian ancestry. This has resulted in significant boost in power to

detect many novel variants for the trait (Variance explained increases from ~7.4% to 11%; additional 145 signals are identified from all-ancestry analysis in addition to European only analyses that identified 935 signals). In addition, they present the first large-scale effort to assess the rare spectrum of variation for AAM utilising gene burden tests by collapsing rare variants which revealed 6 genes associated with AAM. Looking up these genes in voice break data in males is an interesting way suggesting some potentially sex-specific differences. Their GWAS to gene pipeline to map identified signals to genes is comprehensive and utilises multiple data sources to identified in silico functional evidence that colocalise with the signals. Then very interesting and with available data on body weight from age birth to age 8 years, they checked whether the identified 1080 AAM signals had an effect on childhood weight that revealed 3 different trajectories where >50% of AAM signals had no early weight gain trajectory and 44% with moderate weight gain and 1% with high early weight gain. This is very valuable in terms of dissecting those signals with primary effects on puberty timing or early weight gain. In addition to standard pathway enrichment analysis of the identified AAM genes, they have explored an unpublished RNAseq-based differential expression dataset comparing differences in expression between embryonic migration stages of embryonic GnRH neurone in mouse. This works seems very interesting but I think deserves a line on 708 to state relevance of the GnRH neurons to AAM. They show in vitro evidence for interaction between MC3R and GPR83 and also statistical genetic evidence for epistasis between the associated SNPS in both genes. This is very exciting as it enlightens their previous work illustrating MC3R loss of function association with delayed puberty to be enhanced by GPR83 expression. Last but not least, results showing that there are 33 shared signals between AAM and age at menopause and their potential links with later onset conditions life is very informing. This is a huge amount of work shading light on novel mechanisms underlying the roles of genetic determinant of start and end of reproductive lifespan. The methodology and quality of presentation is great. Conclusions are sensible. Publication of this manuscript in Nature Genetics will make sure these reach the wide scientific audience it highly deserves and will benefit.

Response: We thank the reviewer for these comments. As suggested, text has been added to lines 710-714 to describe the role of GnRH neurons in reproduction & puberty.

Some specific comments:

(1) It would be informative to see the allele frequencies of the identified 1080 variants in

each ancestry separately, e.i. in ST2 adding allele frequency for each variants for European ancestry and East Asian ancestry separately.

Response: We have now added the requested allele frequency data to Table S2. As expected, and in line with our previous observations [PMID:29773799], there are some observable differences in frequency across the two ancestry groups. We have added further text to the discussion which describes the potential impact of this and the rationale of our study design (lines 809-813).

(2) Authors may considering estimating potential heterogeneity contributed via differences between the contributing studies (ascertainment (e.g. cancer datasets, biobanks), ancestry (European vs. East Asian)) in the identified genome-wide signals via meta-regression.

Response: In response to this comment, we calculated I^2 values to quantify heterogeneity between the European and East Asian strata (plotted below and added to ST2). Reassuringly, among the 4 European ancestry strata (cancer datasets and biobanks etc.), we observed predominantly very low heterogeneity in signal effects. Plotted below are below I^2 values indicating heterogeneity in effect estimates for each European strata compared to the other 3 European strata combined.

Consistent with expectation, and our previous studies (Horikoshi et al. 2018, PMID: 29773799), we observed far greater heterogeneity between our European and East Asian strata. We note that our strategy in combining these two ancestry groups was to maximise the number of identified loci by increasing our statistical power to identify loci that exhibit homogeneous effects between ancestry groups. Consistent with this, inclusion of the East Asian samples increased the number of identified loci by 145 compared to the European strata alone. As expected, the additional loci identified in this combined ancestry group had low heterogeneity (figure below, right panel) compared to those identified in the European ancestry group alone (left panel). We have added additional text discussing this in lines 804-805 and 809-813.

(3) In exome-wide association results, authors report 6 genes associated significantly with AAM. However, from line 562, they start talking about confirmation of genes from a set of seven. Needs to be clarified where the additional gene is coming from or it a typo.

Response: We apologise for this inconsistency. The 7th gene (*KDM5B*) showed very-near significant association with AAM and was highlighted as it is in the same gene family as *KDM4C*. To avoid confusion, we have now removed mention of this gene in the text.

Referee #2: GWAS, statistical genetics

Ong and colleagues present findings from the largest GWAS of AAM to date, expanding over previous efforts by aggregating European and East Asian ancestry studies. The authors identify more than 1000 signals of association that explain 11% of AAM variance and are associated with delayed and precocious pubertal development in the extremes of the polygenic score distribution. Analyses of exome sequence data highlight genes with a burden of PTV/DMG variants associated with AAM, which accounted for the polygenic risk of common variants in carriers. A total of 660 high-confidence genes were identified using a variant to gene mapping strategy that brought together support from diverse sources, including proximity and colocalization with molecular QTLs. Downstream analyses highlighted the biological complexity of AAM, and highlight clusters of signals that reflect association with early weight gain and BMI later in life.

The manuscript brings together a broad range of diverse analyses and data resources to better understand the genetic contribution to AAM, and represents a considerable advance over previous AAM GWAS efforts. The investigation of the contribution of common and rare variants to polygenic risk was a highlight. The manuscript is generally clearly written and straightforward to follow. My review focuses on the genetic analyses, which are mostly robust, although I do have some concerns and requested clarifications, as outlined below.

- 1. Line 502. Presumably the first four of the five strata are European ancestry. This would be worth emphasizing.*

Response: We have now clarified this on line 506-507.

- 2. Line 511. I could not follow the rationale for signal selection. I can understand using UK Biobank as a reference for LD, but GCTA would normally be used to then identify distinct signals at some pre-defined significance threshold. I think that a genome-wide significance threshold (5×10^{-8}) was used, but it is not clear from the Methods. The criteria for signal selection then seem very complex, based on changes in the magnitude of the P-value after conditioning and R^2 , and I could not see any justification for this. Presumably this is LD r^2 , as opposed to variance explained, but based on UK Biobank European ancestry data? I was then not clear how signals identified in the*

European ancestry meta-analysis from a conditional analysis could be meta-analysed with the East Asian ancestry data (which will not necessarily have been conditioned on the same set of variants). Are the summary statistics from Table ST2 from unconditional or conditional meta-analysis? Are all signals genome-wide significant in conditional analysis?

Response: We apologise that our rationale for signal selection was unclear. Firstly, we can confirm that a significance threshold of $P < 5 \times 10^{-8}$ was used to identify all signals (primary and secondary). Primary signals were identified through a 1Mb distance-based clumping approach. These were then augmented with secondary signals which were a) not in LD with a primary signal ($r^2 < 0.05$), b) did not exhibit more than a 20% change in effect estimate or a change in P-value by more than four orders of magnitude, post conditional analysis.

We appreciate that these practices and thresholds vary across the literature. We note for example that some choose to relax significance thresholds when declaring secondary signals. Based on our experience and consultation with some of the developers of these approaches, we err towards a more conservative approach and choose to report a smaller set of secondary signals, in which we have more confidence. This approach has advantages for some downstream analyses – e.g. Mendelian Randomization is harder to implement using signals that are partially correlated. This is why we filter signals not only on LD, but also the extent to which test statistics change on conditional analysis (as described by Wood *et al*, PMID: 21798870). We apologise that our methods text on signal selection was unclear. This has been clarified.

We did not further meta-analyse signals discovered in the European-only and ancestry combined analyses. Instead, primary and secondary signals were identified in the European-only and ancestry combined analyses, separately. These lists were then amalgamated, removing overlapping signals (based on LD), but without further meta-analysis. The summary statistics in ST2 are from the unconditional meta-analysis.

- 3. Line 511. It wasn't clear how the strategy for signal selection would translate into the all-ancestry meta-analysis. For example, there could be a variant that is in strong LD with a signal identified in the European ancestry meta-analysis based on East Asian ancestry LD, but not European LD (captured in UK Biobank), and it is not clear to me that this would constitute an independent signal. Given that the authors state that their aim is to*

identify signals that are homogeneous across the two strata, I would suggest focussing on clumping genome-wide significant variants from the all-ancestry meta-analysis, requiring independent signals to have $r^2 < 0.05$ in both European and East Asian ancestry reference data. It would recommend NOT referring to the meta-analysis of European and East Asian ancestry data as “all ancestry” as the meta-analysis only brings together data from two ancestries.

Response: We agree with the reviewer that “all ancestry” is not an ideal term given only two ancestry groups. We now describe this throughout as “ancestry combined”.

Our approach for signal selection was to first identify those that were present in Europeans, and then augment this list with additional signals identified in the ancestry combined model. Given the vast majority of samples included here are of European ancestry, we decided not to restrict signal selection to only the combined ancestry group as this would likely miss true European signals that are highly heterogeneous with the East Asian group (informed by our previous observations in Horikoshi *et al*, PMID: 29773799).

As discussed above, we appreciate that different approaches could be taken and there is no consensus in the field. As the reviewer notes, our strategy in combining these two ancestry groups was to maximise the number of identified loci by increasing our statistical power to identify loci that exhibit homogeneous effects between ancestry groups. As a consequence, inclusion of the East Asian samples increased the number of identified loci by 145 compared to the European strata alone. We anticipate that future expanded East Asian focussed analyses will uncover additional loci and provide new insights into allelic heterogeneity at these loci.

We have added comment on these issues in the discussion (lines 809-813).

- 4. Line 520. I would be more interested in understanding evidence for heterogeneity in effects between the European and East Asian ancestry findings. This could be done by comparing effect sizes from the European ancestry meta-analysis with those from the East Asian ancestry meta-analysis. Also of interest is whether the 935 European ancestry signals validate in the East Asian ancestry meta-analysis.*

Response: Of our total 1080 AAM signals, 410 showed at least nominal association among East Asians, including 295 of the 935 signals that were discovered in our European-only sample. As mentioned above, our main aim when adding East Asian ancestry data was to boost power for signals that have homogeneous effects, which led to the addition of 145 loci which we would otherwise have missed.

We calculated I^2 values to quantify heterogeneity between the European and East Asian strata (plotted below and added to ST2):

As expected, and consistent with our previous observations (Horikoshi et al. 2018, PMID: 29773799), we observe high heterogeneity between the European and East Asian samples for signals identified in the European-only strata (left panel). However, the additional signals derived from addition of East Asian samples show predominantly low heterogeneity (right panel), indicating that the addition of East Asian data boosted power to detect signals that are consistent across ancestries. Discussion text outlining this as a limitation has also been added to lines 804-805 and 809-813.

5. *Line 530. The overlap of signals between AAM and AVB in UK Biobank and 23andMe looks convincing. I wonder why the results from these two resources were not meta-analysed to maximise power, and a look-up of effects in the meta-analysis conducted. Whilst the concordance in the direction of effect is interesting, I wonder if the more powerful test would be $P < 0.05$ and concordant direction of effect. Only 2.5% of signals would be expected to meet these two criteria by chance, and I think the support for an enrichment of signals between the two traits would be more convincing.*

Response: We kept the two AVB datasets separate because the traits were measured differently: in years in 23andMe, and in 3-relative categories in UKB. We have now followed the approach in our previous AVB meta-analysis (from Hollis *et al*, PMID: 3221023) where both traits were transformed to (binary) early vs. late AVB and combined using MTAG. We find that 83% of AAM signals are directionally concordant with AVB combined, including 39% at $P < 0.05$. We have now added these results, lines 535-537 and ST4.

6. *Line 562. I was confused what the “seven” genes referred to. Are these the six at exome-wide significance and then KDM5B? I’d be tempted to drop the report of KDM5B since this the near significant findings is actually from a different functional variant category, and was not associated with AVB*

Response: As explained in our response to Reviewer 1, the 7th gene (*KDM5B*) showed very-near significant association with AAM and was highlighted as it is in the same gene family as *KDM4C*. To avoid confusion, we have now removed mention of this gene in the text.

7. *Line 672. The clustering of AAM signals according to early weight gain is very interesting. Have the authors considered testing the effects of these different clusters of SNPs for association with some of the diseases that are associated with adult-onset diseases, by using partitioned polygenic scores, for example?*

Response: This is an interesting suggestion, but it requires extensive analysis and is outside of the scope of this current manuscript.

8. *Line 1006. Many of the signal to gene mappings were based on LD. How was LD assessed, particularly if signals were identified in the multi-ancestry meta-analysis.*

Response: The LD reference we use throughout this manuscript was constructed using data from a random subsample of 25,000 unrelated, European-ancestry UK Biobank participants. We believe this to be adequate for the derivation of LD in our study, given that any dataset we have used is primarily derived from European-ancestry participants (wholly European or ~80% in the case of the ancestry combined meta-analysis). We note that there are few statistical genetics approaches available for handling multi-ancestry LD reference panels.

9. *Line 1032. It would be useful to report the results of the tissue enrichment analysis in the main text, even in just a couple of sentences. It was difficult to dig through the supplementary material to identify the tissues. Presumably, from Supplementary Figure 9, the enriched tissues were adrenal, brain and pituitary?*

Response: This is highlighted in lines 687-690.

10. *Line 1043. What was the justification for the use of PP H4 >0.75 for colocalization from coloc for eQTLs and pQTLs? This doesn't seem to be consistent with what is reported in the main text (PP H4 >0.5)?*

Response: We used the more stringent threshold, H4 PP>0.75, for hypothesis-free genome-wide colocalisation analyses, i.e. within the G2G variant to gene mapping framework to test for colocalisation between GWAS signals and e/pQTLs. For smaller scale, hypothesis-driven analyses, for example colocalisation between proximal signals for traits with clear overlapping aetiology, i.e. between AAM & BMI, we considered the less stringent threshold, H4 PP>0.5, to be appropriate due to the higher prior odds. This is now explained in the Methods.

11. *Line 1068. The calculation of the G2G scores seem rather arbitrary. What was the justification for the different weightings applied across the different sources of evidence?*

Response: The justification of G2G scores is to initially assign similar weights to each separate analysis / line of evidence that connects a variant to a nearby gene. Hence, each has the same maximum arbitrary value (1.5 points).

Referee #3: GWAS, reproductive endocrinology

The study “Understanding the genetic complexity of puberty timing across the allele frequency spectrum” by Kentistou & Kaisinger et al aims to elucidate the biology of pubertal timing by performing multi-ancestry genetic analyses in 800,000 women, complemented by exome sequencing analyses in 220,000 women, in silico variant-to-gene mapping, and integration with gene expression data from mouse embryonic GnRH neurons.

As a result, the authors identify over 1000 independent signals explaining 11% of menarche timing, characterise rare loss of function variants that affect the trait and propose novel biological mechanisms for pubertal timing in females.

Overall, the manuscript is well-written, clear and concise. The work represents another important step forward in understanding the genetic mechanism governing pubertal timing that has also important implications for overall health and wellbeing. It is especially positive to see the authors have included several non-European datasets (the China Kadoorie Biobank, the Biobank Japan, the Korean Genome and Epidemiology Study) to improve diversity in genetic studies.

The selected methods and analyses are appropriate and standard in the field and have mostly been described in sufficient detail to allow replication. The results of these analyses also support the drawn conclusions and nicely informative figures have been used to communicate and summarise the most results.

I was wondering about the conclusion on page 20, lines 832-835 “Together, these insights shed light on mechanisms, including early weight gain and adiposity, hormone secretion and response, and cellular susceptibility to DNA damage, that potentially explain the widely reported relationships between earlier puberty timing and higher risks of later life mortality, metabolic disease, and cancer.”, drawn from the results that show some of the genetic signals shared between menarche and menopause map to DNA damage response genes or to the HPG axis. In the authors’ opinion, regarding the reported associations between puberty timing and cancer risk and in the light on their new results - is the association mostly due to the extended hormonal exposure or due to shared DDR mechanisms (or both)? Could the genetic data somehow be used to test which of these biological mechanisms is driving the association?

Response: We thank the reviewer for this interesting question. Unfortunately, formally testing this remains challenging given our limited ability to systematically map AAM

signals to DDR acting effects. We hope that future work, for example involving the use of cancer biomarkers, will help add more clarity.

Minor questions:

1) Is there any overlap between the breast cancer association consortium and ovarian cancer association consortium samples and rerogen/ukbb/23andme data? Could it possibly affect the results somehow?

Response: We have checked and confirm that there is no known overlap between any of our contributing studies. We cannot preclude the possibility that some individuals may have volunteered to participate in multiple contributing studies. However, this is a generic issue for all population based studies and trying to formally identify such individuals would breach study governance and ethics. Reassuringly we found no evidence of test statistic inflation due to population structure in our analyses (LDSC intercept=1.07, SE=0.03).

2) Were there any ancestry-specific GWAS findings worth highlighting?

Response: Reflecting our sample composition and power, our aim was to highlight signals with robust effects in Europeans-only or with homogenous effects across the two ancestry groups. We are aware of other ongoing ancestry-specific efforts and do not wish to detract from their work. In response to the other reviewers, we have now added information in ST2 on the heterogeneity in effects between the European-only and East Asian strata. We have added text to discuss this aspect of our approach (lines 804-805 and 809-813).

Decision Letter, first revision:

22nd Feb 2024

Dear Dr Ong,

Thank you for submitting your revised manuscript "Understanding the genetic complexity of puberty timing across the allele frequency spectrum" (NG-A62753R). It has now been seen by the original referees and their comments are below. The reviewers find that the paper has improved in revision, and therefore we'll be happy in principle to publish it in Nature Genetics, pending minor revisions to

satisfy the referees' final requests and to comply with our editorial and formatting guidelines.

Sincerely,

Safia Danovi
Editor
Nature Genetics

Reviewer #1 (Remarks to the Author):

Revisions are made to satisfaction. I have no further comments and endorse acceptance of the manuscript for publication.

Reviewer #2 (Remarks to the Author):

The authors have been comprehensive in their responses to my comments and those of the other reviewers. The rationale for methods are now much clearer. I have one remaining issue related to my previous comment 8. I think the use of a European ancestry LD reference for interrogating the results of the ancestry combined meta-analysis is sub-optimal, but I agree there is a lack of methods. I would suggest that the authors highlight this limitation in the methods section (around line 1065). The methods section also still uses "R2" for LD, whilst it should be "r2" and this should be corrected.

Reviewer #3 (Remarks to the Author):

The authors have been responsive to mine and other reviewers' comments, thus I have no further questions/comments and I look forward to seeing this paper published.

Author Rebuttal, first revision:

Response to reviewers' comments

Response: We are grateful to the reviewers for taking the time to review our updated manuscript. Any remaining comments are addressed below.

Reviewer #1: Revisions are made to satisfaction. I have no further comments and endorse acceptance of the manuscript for publication.

--

Reviewer #2: The authors have been comprehensive in their responses to my comments and those of the other reviewers. The rationale for methods are now much clearer. I have one remaining issue related to my previous comment 8. I think the use of a European ancestry LD reference for interrogating the results of the ancestry combined meta-analysis is sub-optimal, but I agree there is a lack of methods. I would suggest that the authors highlight this limitation in the methods section (around line 1065). The methods section also still uses "R2" for LD, whilst it should be "r2" and this should be corrected.

Response: We have added text to the methods discussing the reliance on a European ancestry LD reference as a limitation. All mentions of R2 have been converted to r2.

Reviewer #3: The authors have been responsive to mine and other reviewers' comments, thus I have no further questions/comments and I look forward to seeing this paper published.

--

Final Decision Letter:

13th May 2024

Dear Dr Ong,

I am delighted to say that your manuscript "Understanding the genetic complexity of puberty timing across the allele frequency spectrum" has been accepted for publication in an upcoming issue of Nature Genetics.

Your paper will be published online after we receive your corrections and will appear in print in the next available issue. You can find out your date of online publication by contacting the Nature Press Office (press@nature.com) after sending your e-proof corrections.

Please note that *Nature Genetics* is a Transformative Journal (TJ). Authors may publish their research with us through the traditional subscription access route or make their paper immediately open access through payment of an article-processing charge (APC). Authors will not be required to make a final decision about access to their article until it has been accepted. Find out more about Transformative Journals

Authors may need to take specific actions to achieve compliance with funder and institutional open access mandates. If your research is supported by a funder that requires immediate open access (e.g. according to Plan S principles) then you should select the gold OA route, and we will direct you to the compliant route where possible. For authors selecting the subscription publication route, the journal's standard licensing terms will need to be accepted, including <https://www.nature.com/nature-portfolio/editorial-policies/self-archiving-and-license-to-publish>. Those licensing terms will supersede any other terms that the author or any third party may assert apply to any version of the manuscript.

If you have not already done so, we invite you to upload the step-by-step protocols used in this manuscript to the Protocols Exchange, part of our on-line web resource, natureprotocols.com. If you complete the upload by the time you receive your manuscript proofs, we can insert links in your article that lead directly to the protocol details. Your protocol will be made freely available upon publication of your paper. By participating in natureprotocols.com, you are enabling researchers to more readily reproduce or adapt the methodology you use. [Natureprotocols.com](http://natureprotocols.com) is fully searchable, providing your protocols and paper with increased utility and visibility. Please submit your protocol to <https://protocolexchange.researchsquare.com/>. After entering your nature.com username and password you will need to enter your manuscript number (NG-A62753R1). Further information can be found at <https://www.nature.com/nature-portfolio/editorial-policies/reporting-standards#protocols>

Sincerely,

Safia Danovi, PhD

Senior Editor, Nature Genetics
ORCID: 0009-0007-7822-5479